# Cryo-EM structure of native human uromodulin, a zona pellucida module polymer

Alena Stsiapanava[1,†] (ID), Chenrui Xu[2,3,†] (ID), Martina Brunati[4] (ID), Sara Zamora-Caballero[1] (ID), Céline Schaeffer[4] (ID), Marcel Bokhove[1] (ID), Ling Han[1] (ID), Hans Hebert[1,5] (ID), Marta Carroni[6] (ID), Shigeki Yasumasu[7] (ID), Luca Rampoldi[4] (ID), Bin Wu[2,3,*] (ID) & Luca Jovine[1,2,**] (ID)

## Abstract

Assembly of extracellular filaments and matrices mediating fundamental biological processes such as morphogenesis, hearing, fertilization, and antibacterial defense is driven by a ubiquitous polymerization module known as zona pellucida (ZP) "domain". Despite the conservation of this element from hydra to humans, no detailed information is available on the filamentous conformation of any ZP module protein. Here, we report a cryo-electron microscopy study of uromodulin (UMOD)/Tamm–Horsfall protein, the most abundant protein in human urine and an archetypal ZP module-containing molecule, in its mature homopolymeric state. UMOD forms a one-start helix with an unprecedented 180-degree twist between subunits enfolded by interdomain linkers that have completely reorganized as a result of propeptide dissociation. Lateral interaction between filaments in the urine generates sheets exposing a checkerboard of binding sites to capture uropathogenic bacteria, and UMOD-based models of heteromeric vertebrate egg coat filaments identify a common sperm-binding region at the interface between subunits.

**Keywords** cryo-electron microscopy; polymerization; uromodulin; zona pellucida; ZP domain
**Subject Categories** Microbiology, Virology & Host Pathogen Interaction; Structural Biology; Urogenital System
**The EMBO Journal (2020) 39: e106807**

## Introduction

The ZP "domain" is a conserved sequence of ~ 260 amino acids that was first recognized in UMOD as well as ZP2 and ZP3 components of the mammalian egg coat, the zona pellucida (Bork & Sander, 1992). Functional, biochemical, and structural studies showed that this element is a protein polymerization module that consists of two distinct but topologically related immunoglobulin-like domains, ZP-N and ZP-C (Jovine *et al*, 2002, 2004; Bokhove & Jovine, 2018). These are characterized by different intramolecular disulfide bond patterns and separated by a linker that, in crystal structures of non-polymeric precursor forms of ZP module proteins, can be either flexible or rigid, giving rise to different relative arrangements of ZP-N and ZP-C (Bokhove & Jovine, 2018). For example, the precursor of ZP3 is secreted as an antiparallel homodimer where the two moieties of the ZP module are connected by a largely disordered linker, with each ZP-N domain both lying against the ZP-C domain of the same subunit and interacting with the ZP-C of the other (Han *et al*, 2010). On the other hand, the interdomain linker of the UMOD precursor is entirely structured by forming an α-helix (α1) and a β-strand (β1) that pack against ZP-C; this orients the ZP-N domain so that it homodimerizes with ZP-N from another molecule (Bokhove *et al*, 2016a, 2016b). Despite these differences, in both ZP3 and UMOD the last β-strand of ZP-C (βG)—generally referred to as the external hydrophobic patch (EHP)—is part of a polymerization-blocking C-terminal propeptide (CTP) whose protease-dependent release is required for protein incorporation into filaments (Jovine *et al*, 2004; Schaeffer *et al*, 2009). Notably, in both mammalian egg coat proteins and UMOD, this process is dependent on membrane anchoring of the precursors (Jovine *et al*, 2002; Brunati *et al*, 2015); however, it is unclear how propeptide dissociation triggers polymerization, and the molecular basis of ZP module-mediated protein assembly remains essentially unknown.

To address these questions, we exploited the natural abundance of UMOD (Tamm & Horsfall, 1950; Serafini-Cessi *et al*, 2003) to study ZP module filaments by cryo-electron microscopy (cryo-EM). First recognized as a major component of hyaline casts in 1873

---

1  Department of Biosciences and Nutrition, Karolinska Institutet, Huddinge, Sweden
2  School of Biological Sciences, Nanyang Technological University, Singapore, Singapore
3  NTU Institute of Structural Biology, Nanyang Technological University, Singapore, Singapore
4  Molecular Genetics of Renal Disorders, Division of Genetics and Cell Biology, IRCCS San Raffaele Scientific Institute, Milan, Italy
5  Department of Biomedical Engineering and Health Systems, KTH Royal Institute of Technology, Huddinge, Sweden
6  Department of Biochemistry and Biophysics, Science for Life Laboratory, Stockholm University, Stockholm, Sweden
7  Department of Materials and Life Sciences, Faculty of Science and Technology, Sophia University, Tokyo, Japan
   *Corresponding author. Tel: +65 69082207; E-mail: wubin@ntu.edu.sg
   **Corresponding author. Tel: +46 852480000; E-mail: luca.jovine@ki.se
   †These authors contributed equally to this work

---

   

(Rovida, 1873) and then described as an inhibitor of viral hemagglutination (Tamm & Horsfall, 1950; Serafini-Cessi *et al*, 2003), UMOD is expressed by cells of the thick ascending limb of Henle's loop as a highly glycosylated, intramolecularly disulfide-bonded, and glycosylphosphatidylinositol (GPI)-anchored precursor. This consists of three epidermal growth factor-like domains (EGF I-III), a cysteine-rich domain (D8C), a fourth EGF domain (EGF IV), and the ZP module, followed by a consensus cleavage site (CCS; often referred to as CFCS in other ZP module proteins) and the EHP-including CTP (Fig EV1A) (Serafini-Cessi *et al*, 2003; Bokhove *et al*, 2016a). Hepsin protease-mediated cleavage of the CCS leads to dissociation of mature UMOD from the CTP and triggers its incorporation into homopolymeric filaments (Schaeffer *et al*, 2009; Brunati *et al*, 2015). These protect against urinary tract infections by binding to uropathogenic *E. coli* (UPEC), reduce nephrolithiasis, and are involved in the regulation of water/electrolyte balance and kidney innate immunity (Serafini-Cessi *et al*, 2003; Devuyst *et al*, 2017; Weiss *et al*, 2020). While common variants of UMOD are strongly associated with risk of chronic kidney disease, higher levels of a monomeric form of UMOD that circulates in the serum and regulates renal and systemic oxidative stress were recently linked to a lower risk for mortality and cardiovascular disease in older adults (LaFavers *et al*, 2019; Steubl *et al*, 2020). Thus, elucidating how UMOD polymerization is regulated is not only important for ZP module proteins in general, but also crucial to understand the diverse biological functions of this key urinary molecule.

## Results

### Structure of the UMOD filament

To obtain information on the supramolecular structure of UMOD, we first imaged human urine samples by cryo-EM. This showed that the protein forms semi-regular sheets through lateral interaction of micrometer-long filaments, whose pairing generates features that were previously interpreted as the projection of a double-helical structure (Jovine *et al*, 2002) (Fig 1A). Imaging of purified samples of full-length native UMOD ($UMOD_{fl}$) showed that the majority of filaments had a tree-like structure, resulting from the regular alternation of ~12 nm-long branches protruding at an angle of 50 to 60 degrees from either side of the polymeric core (Fig 1B); other filaments instead adopted a zig-zag shape consistent with early negative stain EM studies of UMOD (Bayer, 1964) (Fig 1C). In agreement with the observation that the two types of structures occasionally interconvert within individual filaments (Fig EV1B), helical reconstruction of $UMOD_{fl}$ showed that these apparently distinct conformations in fact correspond to different views of a single type of filament with 62.5 Å axial rise and 180° twist. The latter parameter, which is even more extreme than the −166.6° helical twist of F-actin (Dominguez & Holmes, 2011), severely complicated structure determination together with the thinness (~35 Å) and flexibility of the filament core. By averaging 288,403 helical segments, we were, however, able to obtain a cryo-EM map of $UMOD_{fl}$ with an estimated average resolution of 3.8 Å, as well as a 3.4 Å map of the corresponding filament core (Figs 1F–J and EV2, Appendix Figs S1 and S2 and Table S1). To inform model building, we also studied

native UMOD digested with elastase ($UMOD_e$), a protease that removes the entire N-terminal region of the protein (EGF I-III + D8C) by cutting at a single site just before EGF IV (Jovine *et al*, 2002) (Fig EV1A and C and Appendix Table S1). This leads to loss of UMOD filament branches (Figs 1D and E, and EV1D), and comparison of the resulting $UMOD_e$ density with that of $UMOD_{fl}$ allowed us to identify the location of the EGF IV N-terminus in the maps (Figs 1F and K, and EV1E). Using this information, we could unambiguously dock the crystallographic model of UMOD EGF IV and ZP-N (Bokhove *et al*, 2016a) into the $UMOD_{fl}$ map and then fit the crystal structure of ZP-C (Bokhove *et al*, 2016a). These placements were validated by the presence of density for the N-glycans attached to ZP-N N396 (Fig 1G and Movie EV1) and ZP-C N513 (Fig EV2B, right panel). Subsequently, a continuous stretch of unexplained density contacting both domains was identified as the ZP-N/ZP-C linker of a third molecule (UMOD 3) that embraces the previously placed ZP-C and ZP-N, which belong to adjacent protein subunits (UMOD 2 and UMOD 4, respectively; Figs 1H and I and 2, Movies EV2 and EV3). This revealed that the relative arrangement of the ZP module moieties of filamentous UMOD is completely different from that of its homodimeric precursor (Bokhove *et al*, 2016a), so that the distance between the centers of mass of the ZP-N and ZP-C β-sandwiches increases from 41 to 91 Å upon polymerization (Fig 3A). Consistent with a ~ 120 Å axial periodicity (Jovine *et al*, 2002), this ZP module conformation allows UMOD monomers to interact head-to-tail (ZP-N-to-ZP-C), with one and two-half subunits per turn (Figs 1F and 2).

Lower map resolution outside the filament core and lack of close structural homologues precluded accurate modeling of D8C. However, different *ab initio* prediction programs suggested that—consistent with the expected presence of multiple intramolecular disulfides (Hamlin & Fish, 1977; Yang *et al*, 2004)—this domain adopts a compact fold with average dimensions that closely match the globular density protruding from EGF IV (Fig EV3A and B). Notably, the density for the short C-terminal tail of hepsin-processed UMOD, whose flexibility is restricted by the last disulfide of the protein ($C_6527$-$C_8582$), merges with that of D8C (Fig EV3B). This suggests that cleavage of UMOD not only activates its ZP-C domain for polymerization, but also allows it to interact with D8C and, in turn, orient the N-terminal region of the protein relative to the core of the polymer. Despite this additional conformational constraint, the latter appears to be highly mobile and swings relative to the filament core, as suggested by the blurred densities seen in 2D class images (Appendix Fig S2C). A multi-body refinement that focused on the branch was then performed to refine the density of the whole N-terminal region of UMOD (Appendix Fig S1 and Fig EV3C). Considering the previous placement of D8C, this local map at moderate resolution agrees with the dimensions of a homology model of EGF I-III (Fig EV3C and D). By combining this information with the refined coordinates of EGF IV + ZP, we could assemble a complete filament model consistent with both the tree and zig-zag views of $UMOD_{fl}$ (Figs 1L and EV3D).

### Filament formation involves a major conformational change in the ZP module's interdomain linker

A dramatic rearrangement of the ZP-N/ZP-C linker region during polymerization underlies the highly different ZP module conformations of the precursor and filamentous forms of UMOD (Fig 3A). In

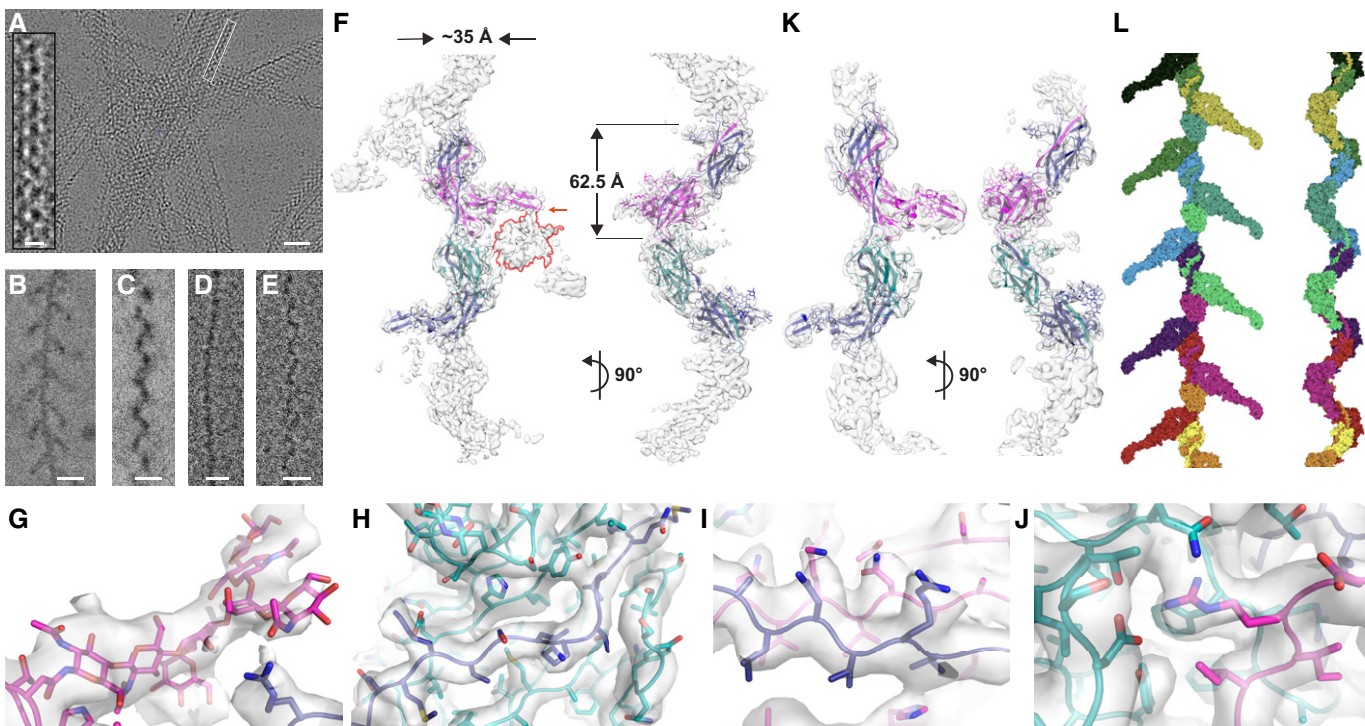

**Figure 1. Overall structure of human UMOD filaments.**

A   Electron micrograph of unstained UMOD filament sheets in human urine. The inset highlights a double helix-like structure resulting from juxtaposition of two individual filaments. Scale bars: 50 nm and 10 nm (inset).

B, C   "Tree" front view and "zig-zag" side view of purified native UMOD filaments, imaged using a Volta phase plate. Scale bars: 10 nm.

D, E   Front and side views of UMOD$_e$ filaments, showing the absence of branches. Scale bars: 10 nm.

F   Orthogonal views of the sharpened cryo-EM map of UMOD$_{fl}$ (3.8 Å resolution), oriented as in panels (B and C), respectively. The map is fitted with an atomic model that consists of a complete EGF IV + ZP module (chain A; blue), the ZP-C domain of a second molecule (chain B; teal), and the EGF IV + ZP-N domain of a third one (chain C; magenta).

G–J   Details of the UMOD$_{fl}$ map shown in panel (F): ZP-N N396 glycan (G); interdomain linker α1β (H) and β1 (I); ZP-C αEFβ/ZP-N βF′ intermolecular interface (J).

K   Sharpened cryo-EM map of UMOD$_e$ (4.0 Å resolution) in two orthogonal views oriented as in panels (D and E). Comparison of this map with that of UMOD$_{fl}$ identifies density belonging to the N-terminal half of UMOD (salmon contour in the front view of panel F), which is lost upon site-specific cleavage by elastase (orange arrow).

L   Goodsell-style depiction of a complete UMOD$_{fl}$ filament model, with protein subunits shown in different colors.

the former, the interdomain linker consists of an α-helix (α1) and a β-strand (β1) that pack against ZP-C β-strand A—an element implicated in polymerization and known as internal hydrophobic patch (IHP; Jovine *et al*, 2004)—and, in the case of β1, also interact with the EHP (Bokhove *et al*, 2016a). In polymeric UMOD, α1 and the amino acids that follow it transform into a 13-residue long twisted β-strand (α1β; alternatively described as two distinct β strands, α1β′ (D430-S434) and α1β″ (A438-M442), separated by a three-residue linker) (Fig 1H and Movie EV2). This substitutes the EHP of the previous subunit by hydrogen bonding to its F and A″ strands as well as facing βA/IHP (Figs 2A and 3B), with linker residue L435 inserted into a conserved hydrophobic pocket formed by IHP M460 and L462, βB V487, and βF L564 (Fig EV4A and B). Interestingly, an identical copy of the DMKVSL sequence that includes α1β′ constitutes β-strand B of UMOD ZP-N (residues 339-344). Antiparallel replacement of EHP/βG by the interdomain linker resembles the donor-strand exchange (DSE) reaction between the subunits of bacterial pili (Waksman, 2017); however, in the case of the ZP module, this β-strand swap is further stabilized by parallel pairing

of the linker β1 region to βG of the ZP-N domain of the following subunit (Fig 1I and Movie EV3). Consistent with the evolutionary conservation of the AG face of the ZP-N domain (Monné *et al*, 2008), this extends the E′FG β-sheet of the latter, creating a surface against which the well-resolved carbohydrate chain attached to N396 packs (Figs 2A and 3B). Notably, the position of this glycan—which is conserved in many ZP module proteins and whose loss in inner ear α-tectorin is associated with human deafness (Sagong *et al*, 2010; Bokhove *et al*, 2016a)—is stabilized by linker β1 itself, which hydrogen bonds to the amide group of a lactosamine GlcNac via the side chain of R449 (Figs 1G and 3B, right panel; Movie EV1). Because of its interaction with both ZP-C and ZP-N, the interdomain linker of polymeric UMOD acts as a molecular belt that links three consecutive protein subunits, burying a total accessible surface area of 1488 Å² (Fig 2).

### Domain–domain interactions in polymeric UMOD

As a result of the structural changes involving the interdomain linker, the ZP-C domain of one subunit (e.g., UMOD 2)

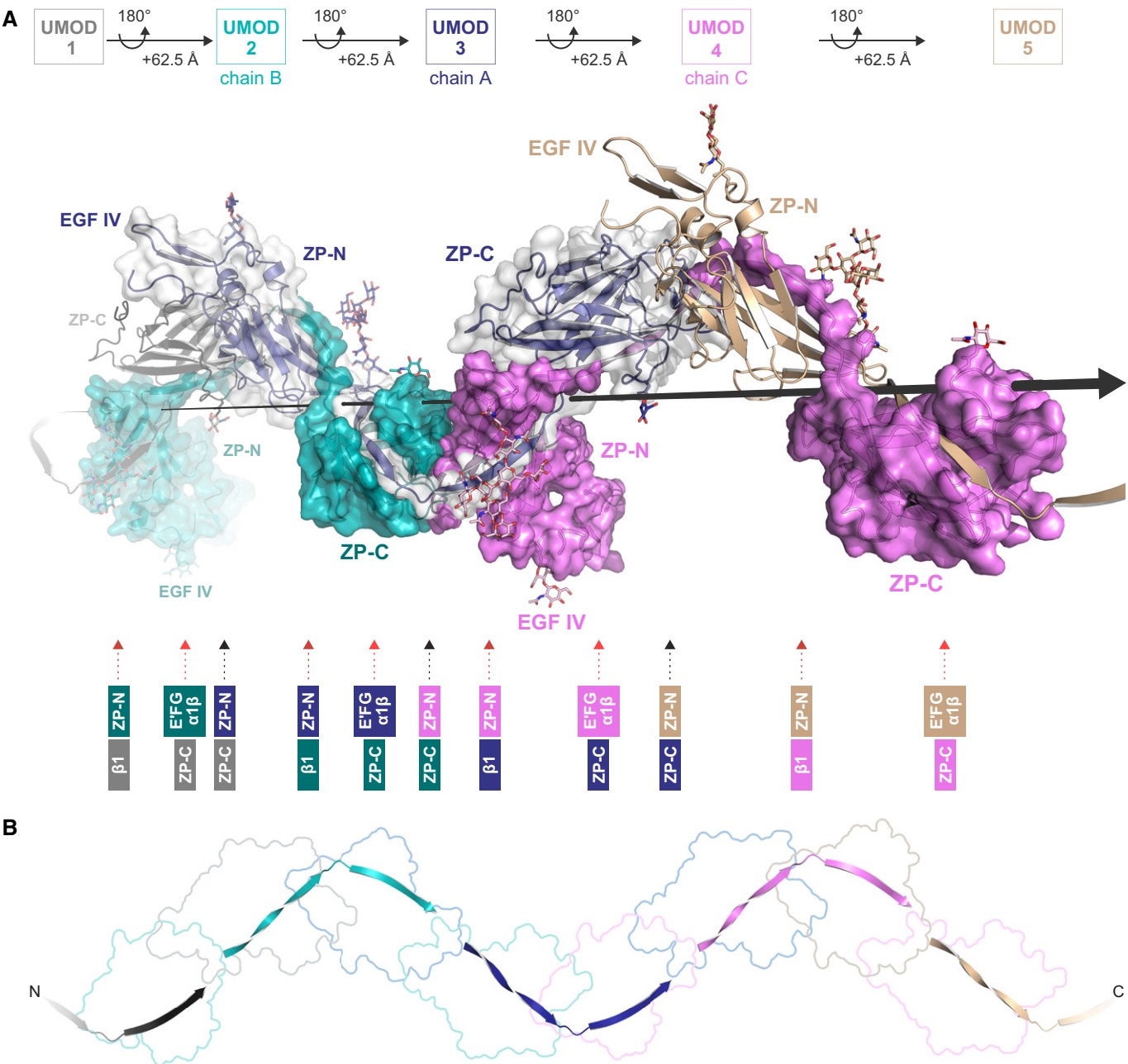

**Figure 2.  Interaction between each copy of UMOD and the ZP modules of four other subunits generates a unique filament architecture.**

A   A section of a UMOD filament is shown that consists of 5 consecutive subunits (UMOD 1–5) related by the helical symmetry operation indicated in the top panel. In the middle panel, where the helical axis is represented by a large black arrow, subunits are depicted in cartoon (UMOD 1, 3, and 5) and surface (UMOD 2, 3, and 4) representation to highlight protein–protein interfaces (with UMOD 1 ZP-N and UMOD 5 ZP-C omitted for clarity). In the filament, the ZP-N/ZP-C linker of each molecule (e.g., UMOD 3) wraps around the ZP-C domain of the subunit that precedes it (UMOD 2) and the ZP-N domain of the subunit that follows it (UMOD 4); additionally, the ZP-N and ZP-C domains of the same molecule are engaged in interactions with the ZP-C domain of the subunit that in turn precedes UMOD 2 (UMOD 1) and the ZP-N domain of the subunit that follows UMOD 4 (UMOD 5), respectively. As summarized in the bottom panel, every UMOD subunit is thus interacting with another four by being engaged in six interfaces that belong to three different types (ZP-N/ZP-C, black arrow; ZP-N/β1, dark red arrow; E′FG, α1β/ZP-C, light red arrow). Subunits 3, 2, and 4 in this figure correspond to Fig 3 chains A, B, and C, respectively.

B   Path of the interdomain linkers of UMOD 1–5, whose domains are outlined in the background. The view is rotated by ~ 40° around the Y-axis, compared to panel (A).

interacts in different ways with two flanking ZP-N domains that belong to the two subsequent subunits (UMOD 3 and UMOD 4) (Fig 2A).

In a first set of contacts, hydrophobic amino acids of UMOD 2 ZP-C, in particular conserved βC F499 and βF L570, interact with residues in the E′FG extension of UMOD 3 ZP-N. These include

conserved L393, the signature Tyr of the ZP-N domain (Y402) and the other near-invariant Tyr of the E'FG extension (Y427), which stacks against highly conserved F456 in the ZP-C IHP (Fig 3B–D and Fig EV4A and C). The E'FG extension is another highly conserved

feature of the ZP-N fold and distinguishes it from other Ig-like domains (Monné *et al*, 2008); together with α1 and the IHP, its signature Tyr has long been implicated in ZP module polymerization (Jovine *et al*, 2004; Monné *et al*, 2008; Schaeffer *et al*, 2009).

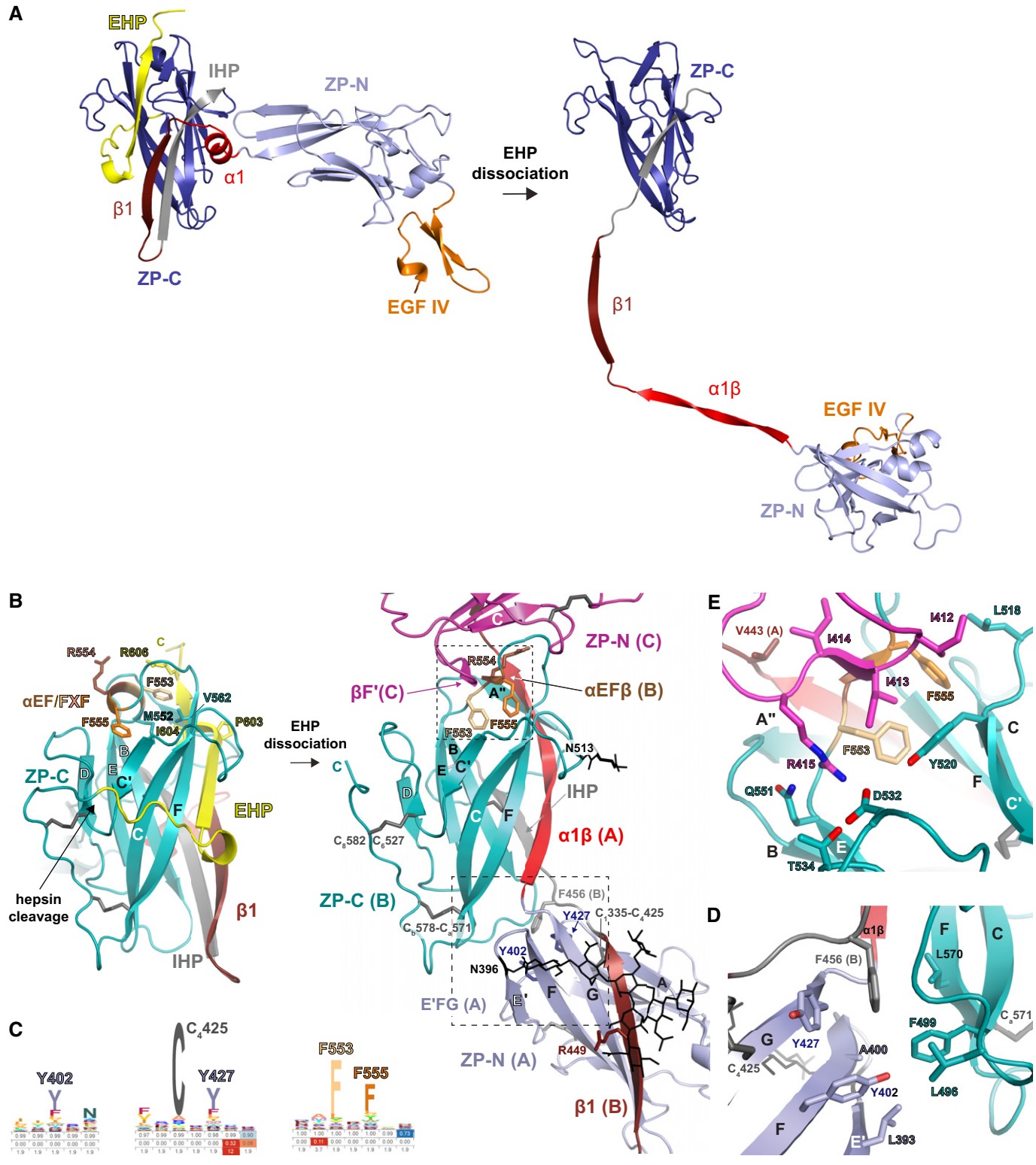

**Figure 3.**

**Figure 3. Conformational changes and protein–protein interactions underlying UMOD polymerization.**

A   Comparison of the precursor and polymeric structures of UMOD shows how dissociation of the EHP triggers a major conformational change in the ZP module. This involves a significant rearrangement of the interdomain linker, which not only completely dissociates from ZP-C but also changes secondary structure upon conversion of α1 in the precursor to α1β in the polymer. Molecules are depicted in cartoon representation, with only one subunit of the UMOD precursor homodimer shown; structural elements are colored as in Fig EV1A, with the N- and C-terminal halves of the ZP-N/ZP-C linker colored bright and dark red, respectively.

B   In the ZP-C domain of the precursor form of UMOD (left panel, teal), the polymerization-blocking EHP β-strand interacts hydrophobically with a short α-helix (αEF) encompassing the FXF motif. Hepsin-mediated cleavage of the CCS of this molecule (chain B/UMOD 2) triggers release of its EHP, which is replaced by α1β from the interdomain linker of a second UMOD subunit (chain A/UMOD 3) (right panel). This allows the FXF motif of molecule B to form an intermolecular β-sheet (αEFβ/βF'; upper dashed box) with the ZP-N fg loop of a third, incoming subunit (chain C/UMOD 4, magenta). Another result of the CCS cleavage is that the C-terminus of mature UMOD 2 is freed for interaction with the D8C domain of the same molecule (not shown). Elements are shown as in panel (A), with disulfide bonds and glycan residues represented by thick dark gray and thin black sticks, respectively; β-strands are labeled as in the UMOD precursor (Bokhove et al, 2016a).

C   Hidden Markov model logos, highlighting the conservation of selected residues shown in panels (B, D, and E).

D   Hydrophobic interactions stabilize the interface between the E'FG extension of the ZP-N domain of chain A and the ZP-C domain of chain B, corresponding to the dashed box in the lower right part of panel (B).

E   Details of the interface between the ZP-C domain of chain B and the ZP-N domain of chain C, showing a different view of the area boxed in the upper right part of panel (B).

The second type of interaction, taking place near the C-terminal end of the α1β strand inserted into the ZP-C domain, involves the ef loop of the latter and the fg loop of the ZP-N domain of the second to next subunit (e.g., UMOD 2 ef/UMOD 4 fg) (Figs 2A and 3B, right panel, and 3E). Remarkably, the ZP-C ef loop contains the highly conserved FXF motif of the ZP module (553-FRF-555; Figs 3C and EV4A and D), which was found to stabilize the homodimeric structure of the precursor of chicken ZP3 (cZP3) by also binding the fg loop of another molecule's ZP-N. This results in the formation of a short intermolecular β-sheet, involving the FXF motif itself and a hydrophobic sequence of the ZP-N fg loop (139-VII-141) of ZP3 (Han et al, 2010). Although UMOD and ZP3 have different relative domain orientations due to fg loop flexibility and alternative oligomerization states (Fig EV4E), the FXF motif of UMOD ZP-C—which in the protein precursor is an α-helix (αEF)—generates an equivalent interface by forming a short β-strand (αEFβ) that pairs with the hydrophobic sequence 412-III-414 in the ZP-N fg loop (βF'; Fig 3B). This is followed by highly conserved R415, which inserts into a negatively charged pocket formed by ZP-C Y520, D532, T534, and Q551 (Figs 1J, 3E, and EV4A, D, and F; Movie EV4). R415 corresponds to ZP3 ZP-N R142, a residue that binds to ZP-C Y243 and D254 (corresponding to UMOD Y520 and D532, respectively) and is essential for homodimerization and secretion of cZP3 (Han et al, 2010). Notably, the helical conformation of the FXF motif in the precursor form of UMOD is stabilized by the EHP; this immediately suggests how hepsin-dependent dissociation of the latter may facilitate the conformational change that activates ZP-C for interaction with ZP-N from another molecule (Fig 3B).

## Head-to-tail mechanism of polymerization

To complement structural data and functionally investigate the mechanism of filament formation, we first expressed full-length UMOD constructs carrying mutations of the ZP-N fg loop/ZP-C ef loop interface in Madin-Darby Canine Kidney (MDCK) cells, which support UMOD secretion and polymerization (Schaeffer et al, 2009). In agreement with the structure, alanine mutation of ZP-N R415 or a 2-residue deletion affecting the ZP-C FXF motif (ΔF555-A556, ΔFA) does not alter UMOD trafficking but completely abolish its polymerization (Figs 4A and EV5A and B). Remarkably, co-transfection

experiments showed that, although neither mutant affects the secretion of wild-type UMOD (wt; Fig EV5C), ΔFA (but not R415A) has a dominant-negative effect on its ability to form filaments (Fig 4B–D). An equivalent result was observed upon co-expression of wt UMOD with a protein variant that cannot polymerize because its CCS has been inactivated (4A; Schaeffer et al, 2009; Figs 4E and EV5C). Considering that the dominant-negative effect of both the ΔFA and 4A mutations is suppressed in R415A/ΔFA and R415A/4A double mutants (Fig 4F and G), we conclude that—consistent with the structural information (Fig 2)—UMOD is a polar filament whose extension depends on the interaction between an activated ZP-C end and the ZP-N domain of an incoming subunit.

A second set of experiments was performed using constructs truncated after the CCS (UMOD-CCS) or the EHP (UMOD-EHP). This showed that, unlike UMOD-CCS which is not secreted but forms intracellular polymer-like structures (Schaeffer et al, 2009), UMOD-CCS R415A or ΔFA are impaired in both secretion and polymerization; on the other hand, wt UMOD-EHP and its R415A or ΔFA variants are secreted and do not form intracellular polymers (Figs 4H and EV5D and E). These results further confirm the functional importance of the ZP-N fg loop and ZP-C FXF motif. At the same time, they suggest that—although the EHP is ultimately replaced by interdomain linker α1β in the context of the UMOD filament (Fig 3B)—this element is crucial for protein secretion even in the presence of polymerization-impairing mutations. Since UMOD-EHP does also not polymerize extracellularly due to lack of cleavage at the CCS (Fig EV5D and F) and wt UMOD only incorporates into filaments attached to the cell from which it was secreted (Fig EV5G), these data also support the idea that EHP dissociation and head-to-tail incorporation into a growing filament are coupled processes occurring at the plasma membrane.

## UMOD homopolymer architecture is conserved in heteropolymeric egg coat filaments

How similar are the filaments of other ZP module proteins—such as those forming vertebrate egg coats—to the UMOD polymer? The ~140 Å structural repeat of mouse egg ZP filaments, thought to consist of heterodimers of ZP2 and ZP3 subunits (Wassarman & Mortillo, 1991), closely matches the helical pitch of the UMOD$_{fl}$ polymer (Fig 1F). Moreover, the structure of the latter agrees with

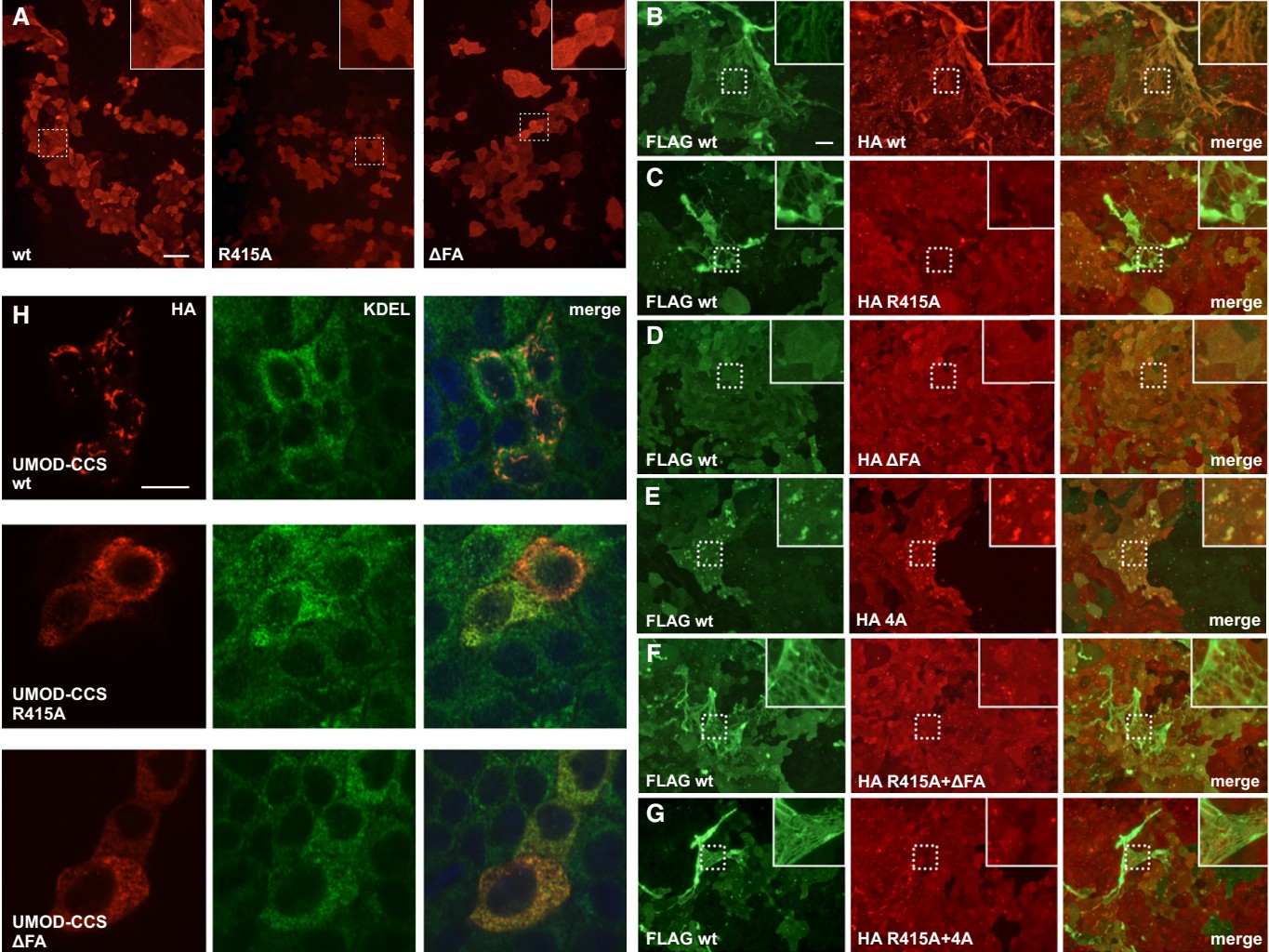

**Figure 4. Effect of polymerization interface mutations on UMOD filament assembly.**

A   Immunofluorescence of unpermeabilized stably transfected MDCK cells shows that, unlike wt full-length UMOD, ZP-N (R415A) or ZP-C (ΔFA) UMOD mutants do not assemble into filaments. Scale bar: 50 μm.

B–G   Immunofluorescence of unpermeabilized cells co-expressing FLAG-tagged wt UMOD (green) and the indicated HA-tagged isoforms (red). UMOD R415A does not incorporate into polymers that only contain wt protein (C). UMOD ΔFA has a dominant-negative effect on wt protein polymerization (D) that is rescued in a double mutant carrying both ZP-N and ZP-C mutations (F). Similarly, the dominant-negative effect of a CCS mutation (4A) that prevents EHP dissociation (E) is suppressed by introducing the R415A mutation in the 4A isoform (G). Scale bar: 50 μm.

H   Immunofluorescence of permeabilized cells expressing soluble isoforms of UMOD truncated before the EHP (UMOD-CCS) shows that wt forms intracellular polymers whereas polymerization interface mutants do not. The intracellular polymers are localized in the endoplasmic reticulum (ER), as shown by co-staining with the KDEL sequence used as ER marker. Scale bar: 10 μm.

the expected solvent exposure of the many glycosylation sites that are variably distributed on the ZP modules of other proteins, including ZP2 and ZP3 (Fig EV4G). Similarly, ZP-N- and ZP-C-based superpositions suggest that the N-terminal repeat region of ZP2 and the C-terminal subdomain of ZP3, which have been implicated in sperm binding (Wassarman & Litscher, 2018) but are dispensable for protein incorporation into the growing ZP (Jovine *et al*, 2002), are compatible with the basic structure of the UMOD filament (Fig EV4H).

To gain additional insights into the organization of heteromeric ZP module filaments, we studied native protein complexes solubilized by digesting the unfertilized egg coats (UFE) of medaka

fish with high and low choriolytic hatching enzymes (HCE/LCE; Yasumasu *et al*, 2010). SDS–PAGE and mass spectrometry (MS) analysis of two fractions of this material separated by size-exclusion chromatography (SEC) (F1-2; Fig 5A) revealed that they both contain a ~ 37 kDa polypeptide encompassing the ZP module of ZI-3 (the medaka homolog of ZP3), as well as ~16 and ~18 kDa fragments corresponding to the ZP-N and ZP-C domains of ZI-1,2 (a subunit that replaces ZP2 in the fish egg coat) (Fig 5B, D and E). Consistent with their different native-PAGE profiles (Fig 5C), native MS of chemically cross-linked F1 and F2 revealed that the latter is a heterocomplex of the three aforementioned species (Fig 5F), whereas the former contains monomers, dimers, and

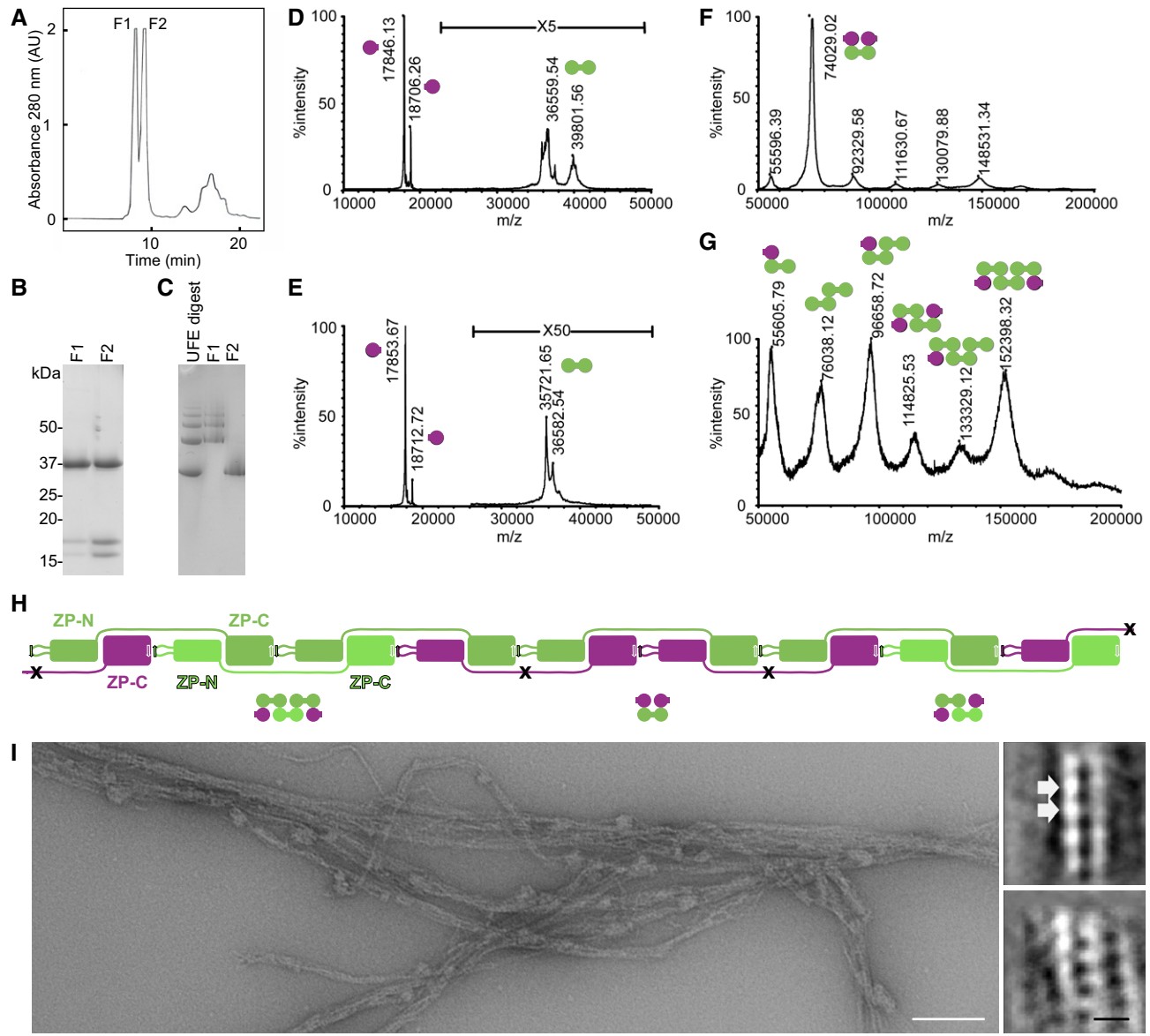

**Figure 5. Egg coat filament organization is consistent with the architecture of UMOD.**

A Analytical SEC of UFE HCE/LCE digests produces two protein peaks, F1 and F2.

B, C Reducing SDS–PAGE (B) and native PAGE (C) of SEC-purified UFE digests with indicated molecular weight markers. The different SEC elution volumes of F1 and F2 (A) reflect different levels of egg coat filament digestion by HCE/LCE.

D, E TOF-MS spectrograms of purified F2 (D) and F1 (E) products. Round-shaped symbols near the peaks indicate the two moieties of the intact ZP module of ZI-3 (green) or the separated ZP-N and ZP-C domains of LCE-cleaved ZI-1,2 (purple).

F, G TOF-MS spectrograms of cross-linked products of F2 (F) and F1 (G), with domain symbols indicating the deduced subunit composition of cross-linked products.

H Schematic representation of subunit interactions in medaka egg coat filaments. While ZI-1,2 subunits (purple) incorporate into the filaments upon activation of monomeric precursors (Appendix Fig S3A), the variable oligomeric state of the ZI-3 precursors (Appendix Fig S3B) allows this subunit to be incorporated in either dimeric (green) or monomeric (light green) form. Digestion of the resulting polymers through specific cleavage of the ZI-1,2 interdomain linker by LCE (black crosses) would solubilize filament fragments corresponding to the macromolecular complexes identified in (F) and (G).

I Electron micrograph of negatively stained medaka egg coat filaments. On the right are representative 2D classes, indicating a separation of ~ 65 Å between repeating units within the polymers (white arrows). Scale bars: 100 nm (left panel) and 10 nm (bottom right panel).

trimers of ZI-3 bound to either one or two separate copies of ZI-1,2 ZP-N/ZP-C (Fig 5G). This agrees with the observation that LCE cleaves the interdomain linker of the ZP module of ZI-1,2 (but not that of ZI-3) (Yasumasu *et al*, 2010), processing a site that corresponds to the short loop between UMOD α1β and β1 (Fig 5H).

Moreover, it is consistent with SEC-multi angle light scattering (MALS) evidence that the precursors of ZI-1,2 and ZI-3 are monomeric and monomeric/dimeric, respectively (Appendix Fig S3). Together with the finding that the polymeric core of medaka UFE consists of repeating units spaced by ~65 Å (Fig 5I), these results

strongly suggest that vertebrate egg coat filaments share the same basic architecture as UMOD.

## Supramolecular organization of UMOD filament sheets

Incubation of purified UMOD with a NaCl concentration comparable with that of urine induces the formation of sheets (Fig 6A, top) similar to those shown in Fig 1A. Correlation averaging analysis of this negatively stained material detects the presence of a two-fold axis perpendicular to the plane of the sheets (Fig 6A, bottom), indicating an antiparallel filament arrangement. By combining this information with the composite map of UMOD$_{fl}$ (Fig EV3D), we could build a UMOD sheet model (Fig 6B) that readily explains the apparent double-helical features of the native protein (Fig 1A, inset) and, based on the location of ZP module N-glycans, accounts for the diffuse density observed in the space between filaments (Fig 6C).

# Discussion

The first detailed structure of a polymeric ZP module protein reveals a unique filament architecture whose features, to the best of our knowledge, do not resemble those of any previously reported biological polymer. In particular, the structural information reported in this study clearly shows that, unlike hypothesized in the case of egg coat proteins (Jovine *et al*, 2006; Egge *et al*, 2015; Okumura *et al*, 2015; Louros *et al*, 2016), UMOD neither polymerizes by forming an amyloid cross-β structure nor assembles through contacts that mostly involve the ZP-N domain or, alternatively, ZP-C. Instead, an unprecedented interlocked configuration of ZP-N and ZP-C domains forms the basis of the filament core. Notably, our data also do not support the UMOD architecture suggested by a recent tomographic analysis, where polymers are made up of ZP modules that simply stack in a zig-zag manner, without any conformational change of the interdomain linker and associated separation of ZP-N and ZP-C (Weiss *et al*, 2020). On the other hand, our results are in agreement with a study of the UMOD filament core (Stanisich *et al*, 2020) that appeared while this work was under revision and publicly available as a preprint version (preprint: Stsiapanava *et al*, 2020).

Together with our mutagenesis and biochemical experiments, the cryo-EM structure of filamentous UMOD has long-ranging implications for different aspects of ZP module matrix assembly and function (Figs 6 and 7).

## Interlocking mechanism of UMOD ZP module polymerization

By combining previous crystallographic information on the homodimeric precursor of UMOD (Bokhove *et al*, 2016a) with the present cryo-EM structure of its filament (Figs 1 and 2) and associated mutagenesis experiments (Figs 4 and EV5), a four-step mechanism of ZP module-mediated protein polymerization can be proposed (Fig 7 and Movie EV5). Consistent with the membrane-anchoring requirement for UMOD incorporation into filaments, the process starts when type II transmembrane serine protease hepsin cleaves the CCS of GPI-anchored UMOD precursors (Brunati *et al*, 2015), which in Fig 7 are represented by two homodimers (UMOD 1

(gray)/UMOD 2 (teal) and UMOD 3 (blue)/UMOD 4 (magenta)) corresponding to the assembled subunits shown in Fig 2. As a result of the orientation of the UMOD homodimer on the membrane and its intrinsic asymmetry (Bokhove *et al*, 2016a), only one of the moieties of each precursor (UMOD 2 and UMOD 4, respectively) is initially cleaved (step I; Fig 7A). Pulling of the CCS inside the deep specificity pocket of the enzyme, as well as recognition of the substrate's prime residues (Barré *et al*, 2014), dislodges the EHP from the corresponding ZP-C domain. This activates the latter for polymerization by releasing its αEF helix and allowing it to engage in an antiparallel β-sheet interaction (αEFβ/βF′) with the fg loop of a ZP-N domain from another dimer (Step II; Figs 3B and E, and 7B). Notably, pairing of the ZP-N domains of the homodimeric UMOD precursor forms an extended β-sandwich, oriented so that one ZP-N copy (UMOD 2 and UMOD 4 in Fig 7A and B) sits on top of the other relative to the plasma membrane (Bokhove *et al*, 2016a). This allows the UMOD branch preceding each ZP-N to project laterally and lie flat on the surface of the membrane; at the same time, it positions the top ZP-N in a favorable position to attack the activated ZP-C of a nearby half-cleaved precursor. Because the top ZP-N copy corresponds to the ZP-C domain that is preferentially cleaved within its own homodimer, interaction between half-cleaved precursors generates a protofilament that remains membrane-bound via the non-cleaved moieties of each homodimer (UMOD 1 and UMOD 3) (Step III; Fig 7C). Importantly, the EHP of the UMOD precursor faces the β1 strand of the interdomain linker, which is paired to ZP-C IHP; detachment of the EHP thus not only affects the αEF helix, but—by loosening the interaction between β1 and ZP-C—also facilitates the conformational change of the interdomain linker and structural transformation of its α1 helix region (Figs 3A and 7B). This is a prerequisite for the final step of assembly, where reorientation of the intact subunit of each incorporated homodimer (UMOD 1 and UMOD 3) as a result of protofilament formation allows it to also be cleaved by hepsin (Fig 7C), locally detaching the nascent polymer from the membrane (Fig 7D). At this point, sequential or concerted conformational changes in the interdomain linkers of the two moieties of each incorporated homodimer allow the newly cleaved subunit (e.g., UMOD 3) to wrap around the other (UMOD 4), replacing the UMOD 4 ZP-N/UMOD 3 ZP-N interface of the precursor with the UMOD 4 ZP-N/UMOD 3 β1 interaction observed in the filament (Fig 2). This frees the ZP-N of UMOD 3 so that it can engage in its own αEFβ/βF′ interaction with the activated ZP-C domain of UMOD 1 (Fig 2A). At the same time, reminiscent of DSE between the subunits of bacterial pili (Waksman, 2017), the α1β linker region of UMOD 3 fills the open G strand/EHP groove of the activated UMOD 2 ZP-C, facing its IHP (Fig 3B); similarly, the activated ZP-C of UMOD 3 itself is completed by α1β of the UMOD 4 interdomain linker. In both cases, these interactions are stabilized by packing of the ZP-N E′FG extension of one subunit against the ZP-C FC end of the previous (Fig 3B and D).

The head-to-tail directionality of the proposed polymerization mechanism explains the dominant-negative effect of the ΔFA mutation (Fig 4) and, as discussed below, pathologic mutations in human ZP module proteins. By postulating that the protofilament remains membrane anchored, it also rationalizes the observation that activated UMOD subunits do not incorporate into polymers growing on the surface of nearby cells (Fig EV5G). Importantly, this provides a solution to the physical problem of assembling the long

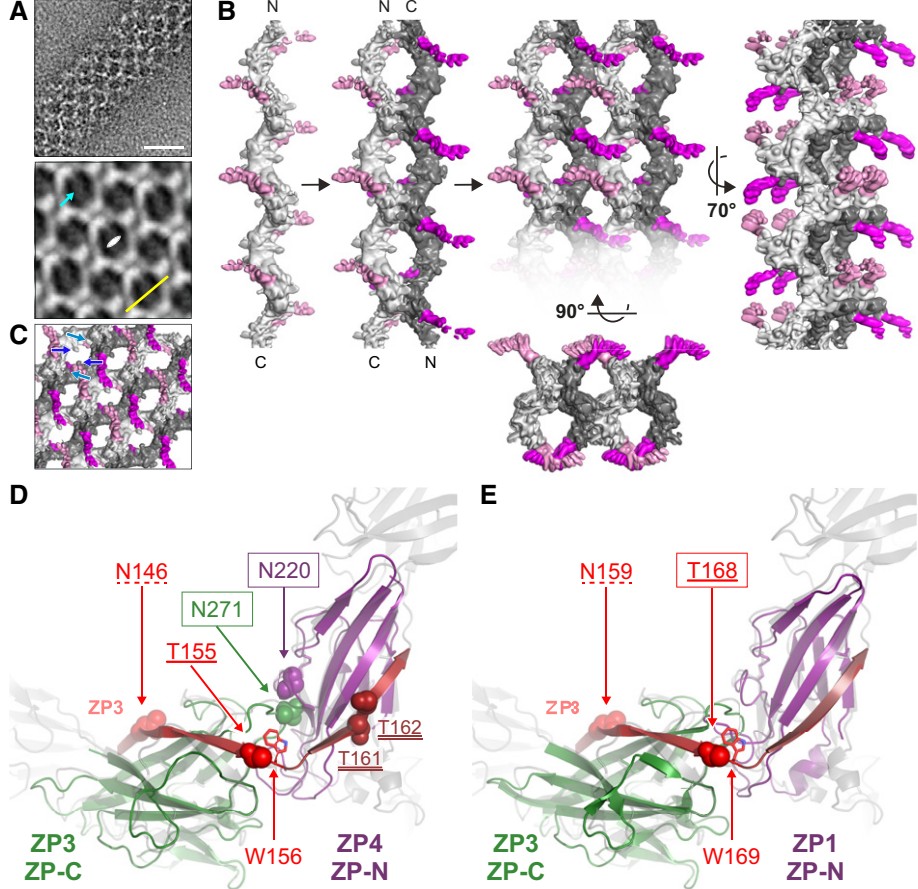

**Figure 6.   Modeling of UMOD sheet architecture and ZP module filament binding sites for UPEC and sperm.**

A   Negative stain micrograph of a sheet formed by purified UMOD filaments in 100 mM NaCl (top). Analysis of this material by correlation averaging reveals a periodic pattern (bottom) with a two-fold symmetry axis perpendicular to the plane of the sheet (white symbol). Scale bars: 250 Å (top) and 125 Å (bottom).

B   UMOD sheet model, generated by antiparallel juxtaposition of multiple copies of the UMOD$_{fl}$ composite map, according to the 2D information shown in panel (A). Due to lack of information in the direction perpendicular to the plane of the sheets, the exact position of adjacent filaments relative to this plane is unknown; for simplicity, the depicted model has been generated by assuming that the cores of the filaments making up the sheet lie on the same plane. Individual filaments are alternately colored light and dark gray, with N-terminal protein regions highlighted in pink and magenta, respectively. The zig-zag view of two adjacent filaments in the sheet resembles the projection of a double helix (see inset of Fig 1A), and the regular presentation of UMOD branches generates a molecular Velcro surface for the capture of UPEC.

C   Comparison of the UMOD sheet model with the bottom part of panel (A) suggests that the diffuse density protruding into the space between adjacent filaments (cyan arrow) corresponds to the glycan(s) attached to N396 and/or N513 (light and dark blue arrows, respectively).

D   Detail of a porcine ZP filament model, generated by superposition of homology models of adjacent ZP3 and ZP4 subunits (green and purple, respectively) onto the structure of UMOD$_{fl}$ (semi-transparent gray), and projection of the residues of the interdomain linker of another ZP3 subunit onto the UMOD ZP-N/ZP-C linker (red). A cluster of experimentally verified glycosylation sites is shown, with side chain atoms depicted as spheres and boxes highlighting the proximity of ZP3 N271 and ZP4 N220, whose N-glycans mediate sperm binding in pig (Yonezawa, 2014). Note how these residues are also close to the conserved N- (dashed underline) and O- (site 1, single underline; site 2, double underline) glycosylation sites in the ZP3 interdomain linker. The invariant ZP3 Trp that follows site 1 is shown in stick representation at the interface with ZP4.

E   Detail of the ZP3/ZP1 interface of an avian egg coat filament model, assembled and represented as described for panel (D). The box highlights the location of O-glycosylation site 1, important for *in vitro* binding of chicken ZP3 to sperm (Han et al, 2010).

UMOD polymer in an extracellular environment that is constantly subjected to high flow; clarifies why the growing mammalian ZP thickens from the inside, a process that also depends on membrane anchoring of ZP2 and ZP3 (Qi *et al*, 2002; Jovine *et al*, 2002); and is compatible with the recent hypothesis that membrane tethering of ZP module protein α-tectorin is essential for generating layers of extracellular matrix whose progressive release generates the tectorial membrane (Kim *et al*, 2019a).

Conservation of the structural elements underlying the mechanism, including the length of the interdomain linkers (~22-26 residues in ZP1-3 compared to 24 residues in UMOD), suggests that other members of the ZP module protein family use a similar mechanism to assemble into filaments that share a common basic architecture; indeed, unusual ZP module proteins with minimal interdomain linkers, such as endoglin and betaglycan, do not form polymers (Bokhove & Jovine, 2018). At the same time, the fact that

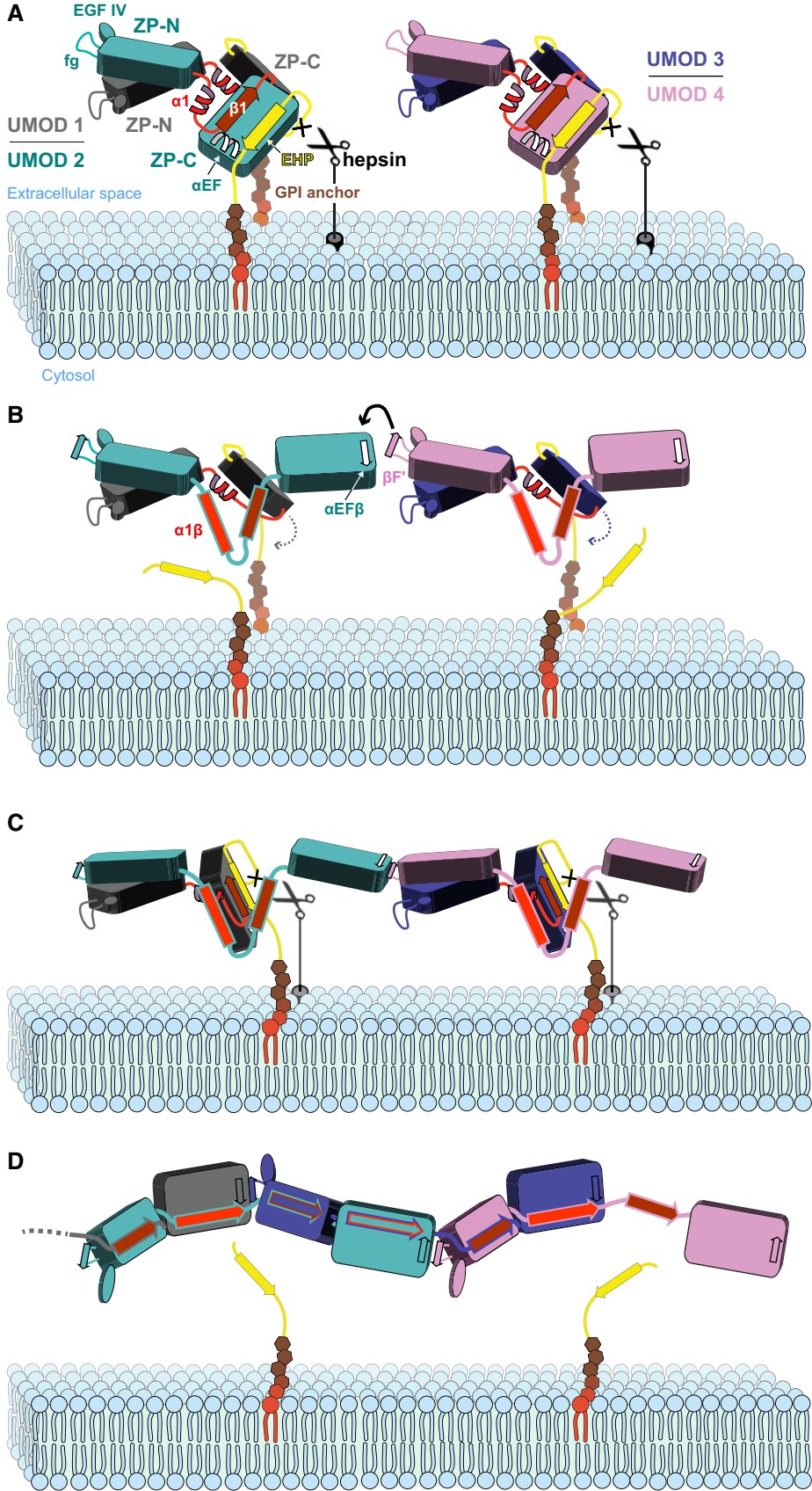

**Figure 7.**

**Figure 7. Proposed mechanism of UMOD polymerization.**

A   Hepsin cleaves the membrane-proximal CCS sequence of GPI-anchored UMOD homodimers, triggering the dissociation of the EHP from the corresponding ZP-C domain. For clarity, UMOD branch domains preceding EGF IV have been omitted.

B   EHP displacement activates ZP-C for polymerization by allowing it to form an intermolecular β-sheet with an incoming ZP-N domain from another homodimer.

C   Reorientation of the second ZP-C allows it to also be processed by hepsin, locally detaching the growing filament from the membrane.

D   The interdomain linker, whose ordered precursor structure is also perturbed upon EHP dissociation, undergoes a major rearrangement that both replaces the ZP-N/ZP-N interface of the UMOD precursor and compensates for the loss of the EHP by DSE, completing filament formation.

ZP3 precursors can already contain cross-subunit αEFβ/βF′ interactions (Han *et al*, 2010) may facilitate their incorporation as dimers in the egg coat of species such as chicken (Han *et al*, 2010), humans (Zhao *et al*, 2004), and fish (Fig 5). In these and other heteropolymeric systems, the variable sequences of the interdomain linkers of different components are also expected to play a major role in determining how subunits incorporate into filaments (Suzuki *et al*, 2015).

## Functional implications: from antibacterial defense to fertilization

The structure of polymeric UMOD provides an essential framework to help understanding the biology of this important urinary protein (Serafini-Cessi *et al*, 2003; Devuyst *et al*, 2017), as well as all other members of the large superfamily of ZP module-containing extracellular molecules (Litscher & Wassarman, 2015). In particular, our analysis of UMOD filament sheets (Fig 6A–C) immediately suggests how the supramolecular organization of the protein allows it to trap UPEC. This is because lateral pairing of multiple UMOD filaments generates a surface whose faces expose checkerboard-like arrays of EGF I–III + D8C regions (Fig 6B), each carrying a copy of the high-mannose glycan recognized by type I pilus adhesin FimH (van Rooijen *et al*, 1999; Pak *et al*, 2001). A similar strategy is likely to be employed by the major receptor for FimH-positive bacteria in the gastrointestinal tract, glycoprotein 2 (GP2) (Hase *et al*, 2009; Kolenda *et al*, 2018), to counteract infection by type I-piliated *Escherichia* and *Salmonella* strains. This is because, in agreement with the idea that their genes originated by duplication, GP2 is structurally very similar to UMOD by containing a D8C domain N-terminal to its ZP module (Kobayashi *et al*, 2004; Yang *et al*, 2004). Notably, egg coat filaments can also interact laterally to form bundles (Fig 5I). Because the N-terminal region of ZP module proteins (whose dimensions can significantly exceed those of the ZP module itself) must rotate around the filament core together with ZP-N during polymerization (Fig 7 and Movie EV5), and considering that not all ZP module protein precursors contain membrane-anchoring elements (Litscher & Wassarman, 2015), such interactions are likely to occur when filaments are at least partially formed (and, in the case of UMOD, detached from the membrane), exposing their branches in a regular orientation that is compatible with lateral pairing of the polymers.

In yet another part of the body, exposure of functionally important regions of ZP2 and ZP3 subunits adjacent to each other within mammalian egg coat filaments could allow them to cooperate in mediating the initial attachment of gametes at fertilization. Interestingly, modeling of the N-terminal ZP-N repeat region of mammalian ZP2 with the same orientation relative to the filament axis as UMOD's branch suggests that the second repeat of ZP2—whose

post-fertilization cleavage by cortical granule protease ovastacin is crucial to establish a definitive block to polyspermy (Burkart *et al*, 2012)—could in principle contact the conserved ZP-C subdomain of ZP3 (Fig EV4H). This potential interaction might affect sperm binding by influencing the efficiency of ZP2 cleavage and, like in the case of the D8C domain of UMOD and its C-terminus (Fig EV3B), could depend on the C-terminus of mature ZP3 being freed upon CFCS processing and incorporation into a filament. The ZP-C subdomain of ZP3 could also function by affecting the presentation of the sperm-binding domain of ZP2, or, alternatively, the physical proximity of the subunits may allow them to generate a hybrid sperm-recognition surface that includes regions from both proteins.

Mapping of egg coat protein sites implicated in sperm recognition in mammal and bird onto UMOD-based models of the respective filaments strongly supports the latter concept, which also involves the interdomain linker of the ZP module itself—a region whose presentation within the filament completely depends on wrapping around two other protein molecules (Fig 2). A particularly relevant example are two O-glycosylation sites conserved within the ZP-N/ZP-C linker of mammalian ZP3 (Chalabi *et al*, 2006), the first one of which is also modified in chicken where it was shown to be important for sperm binding (Han *et al*, 2010). Projection of these sites onto the structure of polymeric UMOD suggests that site 1 is exposed at the end of α1β in the interdomain linker of ZP3, whereas site 2, which encompasses two or three closely spaced Ser/Thr residues depending on the species, is located within its β1. This periodically positions the O-glycosylation sites around the interface between ZP3 ZP-C and the ZP-N domain of the subsequent subunit (ZP1/2/4 in mammals or ZP1 in chicken), whose interaction most likely involves the invariant Trp of ZP3 that immediately follows site 1 and corresponds to V495 at the interface between UMOD ZP-C and ZP-N (Fig 6D and E). Remarkably, the same egg coat filament region is implicated by studies of gamete interactions in pig and bovine, where ZP subunit ZP4 plays a key role in fertilization by forming a sperm-binding complex together with ZP3 (Kanai *et al*, 2007; Yonezawa *et al*, 2012). The activity of this heterocomplex largely depends on tri- and tetra-antennary carbohydrate chains attached to pig ZP4 N220 (Yonezawa, 2014), and it is striking that this residue is predicted to be located just next to N271—the only ZP3 sequon that carries the same two types of glycans (Yonezawa *et al*, 1999)—as well as in close proximity to ZP3 O-glycosylation sites 1 and 2 (Fig 6D). The latter may also contribute to sperm binding in the pig (Yurewicz *et al*, 1991), and—together with the near-invariant sequon of ZP3 at the beginning of α1β—additional glycosylation sites located in the same region may regulate the species specificity of gamete interaction in other organisms.

Taken together, these considerations provide further support for the general idea that heteropolymeric ZP module protein filaments

have the same basic architecture as UMOD; moreover, they strongly suggest that carbohydrate chains attached to adjacent ZP subunits wrapped by the ZP3 interdomain linker contribute to a common sperm-binding interface on the surface of egg coat filaments. Consistent with the aforementioned studies on pig and bovine ZP3/ZP4 (Kanai *et al*, 2007; Yonezawa *et al*, 2012), the difficulty to fully recapitulate such a complex system using non-interacting forms of ZP subunits expressed *in vitro* may have significantly hampered the decade-long search for a bona fide sperm counterpart of mouse and human ZP3.

### Interpretation of pathologic mutations in human ZP module proteins

Because the conformations of the precursor and polymeric forms of ZP module-containing proteins are so different, the availability of structural information on how these molecules assemble into polymers will make it possible to more comprehensively assess the effect of missense mutations linked to human diseases.

For example, the structure of UMOD rationalizes the dominant-negative effect of hearing loss-associated *TECTA* mutation Y1870C (Legan *et al*, 2005), which affects one of the two highly conserved Tyr residues in the ZP-N E′FG extension of human α-tectorin, by showing that the corresponding residue in UMOD (Y402) is part of the interface with the ZP-C domain of the adjacent molecule (Fig 3B, right panel, and C, D; Fig EV4C). This suggests that the mutation disrupts the tectorial membrane not only by interfering with formation of the invariant $C_1$-$C_4$ disulfide of ZP-N (Monné *et al*, 2008), but also by directly affecting one of the polymerization interfaces of α-tectorin. Moreover, comparison of the location of this TECTA ZP-N mutation to that of UMOD ZP-N R415A (Fig 4) in light of the proposed mechanism of polymerization (Fig 7) explains why only the former has a dominant-negative effect on polymerization.

A different but related example of dominant-negative mutation is represented by recently described human ZP3 A134T, which causes female sterility due to empty follicle syndrome (Chen *et al*, 2017) and affects a residue at the N-terminal end of ZP-N βG. Due to the peripheral location of this strand in the ZP-N β-sandwich, the mutation is not expected to compromise the folding of the ZP3 precursor (Monné *et al*, 2008; Han *et al*, 2010); however, βG has a crucial role in polymerization by pairing with interdomain linker β1 from another subunit (Fig 3B, right panel). This clearly suggests that introduction of a polar side chain in βG will prevent the extension of the ZP-N hydrophobic core by β1, thus interfering with ZP filament formation and ultimately leading to lack of a ZP and oocyte degeneration.

### Regulation of ZP module filament polymerization

Perhaps mimicking the sperm-binding ZP-N domain repeats found in the N-terminal region of egg coat proteins from both vertebrates and invertebrates, such as ZP2 and VERL (Callebaut *et al*, 2007; Monné *et al*, 2008; Raj *et al*, 2017), stacking of isolated ZP-N domains can generate filamentous structures *in vitro* (Jovine *et al*, 2006). However, in agreement with the presence of both domains in the C-terminal region of all polymeric ZP module proteins (Bork & Sander, 1992; Litscher &

Wassarman, 2015), the cryo-EM structure of UMOD conclusively shows that both ZP-N and ZP-C are essential for filament formation (Fig 2). This observation raises the question of what is the biological significance of the isolated ZP-N domains found in a number of non-egg coat proteins from worm to humans (Jovine *et al*, 2006). Although further experimental studies will clearly be required to address this issue, our structural and mutagenesis data raise the intriguing possibility that these molecules may regulate the polymerization of full-ZP module proteins by attaching themselves to the growing end of their filaments. This is because, reminiscent of the dominant-negative effect of mutant ZP-C and of the pathogenic ZP-N mutations discussed in previous sections, such ZP-N-only subunits would effectively block further polymer extension due to the lack of a ZP-C domain. Interestingly, a low-abundance truncated form of uromodulin that lacks ZP-C was identified in human urine, suggesting that regulation of polymer extension could also be determined by differential processing that generates a ZP-N-only variant of the same protein (Micanovic *et al*, 2018).

### Conclusions and outlook

The interlocked structure of the UMOD ZP module polymer, where 25% of the surface area of each molecule is buried in interactions with four other subunits, makes it highly stable to proteolysis (Fig EV1C–E) as well as remarkably resistant to chemical denaturation (Fig EV1F). These properties, which result from the significant reinforcement of subunit–subunit interactions by the semi-continuous wrapping of interdomain linkers around the filament core (Fig 2), can be further strengthened by covalent cross-linking in other ZP module proteins, as observed, for example, in the case of egg coat subunits from fish to humans (Greve & Wassarman, 1985; Yasumasu *et al*, 2010; Nishimura *et al*, 2019) or insoluble cuticlin components of the nematode larval alae (Sapio *et al*, 2005). These considerations explain why the ZP module has been evolutionarily selected for assembling a variety of protective matrices (Litscher & Wassarman, 2015). Future structural studies of filaments made by other members of this large family of mosaic extracellular molecules will clarify how the interplay between a common ZP module and variable N-terminal domains mediates such a wide range of essential biological functions.

## Materials and Methods

### Purification and limited proteolysis of native human UMOD filaments

Full-length UMOD filaments (UMOD$_{fl}$) were purified from the urine of a healthy human donor using the diatomaceous earth method (Serafini-Cessi *et al*, 1989), dialyzed against Milli-Q water, and concentrated to 4 mg ml$^{-1}$ using Amicon Ultra Centrifugal Filter Units with Ultracel-50K (Merck Millipore). UMOD$_e$ filaments were obtained by limited digestion of UMOD$_{fl}$ with elastase (Jovine *et al*, 2002), dialyzed overnight at 4°C against 10 mM Na-HEPES pH 7.0, and concentrated to 1 mg ml$^{-1}$. Both UMOD$_{fl}$ and UMOD$_e$ samples were flash-frozen in liquid nitrogen and stored at −80°C until further use.

## Cryo-EM data collection

UMOD$_{fl}$ (0.85 mg ml$^{-1}$), UMOD$_e$ (1 mg ml$^{-1}$), and urine were applied in 3 µl volumes onto glow-discharged Quantifoil Au R2/2 holey carbon 300 mesh grids (Quantifoil). Grids were blotted for 2.0 s and plunged into liquid ethane cooled by liquid nitrogen, using a Vitrobot Mark IV (Thermo Fisher Scientific).

All cryo-EM experiments were performed at the Cryo-EM Swedish National Facility, SciLifeLab, Stockholm. Movies for UMOD$_{fl}$ or UMOD$_e$ were collected with the EPU data acquisition software on a Titan Krios electron microscope (Thermo Fisher Scientific) operated at 300 kV, using a Gatan K2 Summit direct electron detector coupled with a Bioquantum energy filter with 20 eV slit. The defocus range for UMOD$_{fl}$ was between −1.5 and −3.5 µm, the pixel size was 1.06 Å/pixel, and the total dose was ~40 electrons/Å$^2$, distributed in 40 frames. For UMOD$_e$, the defocus range was kept between −1.4 and −3.0 µm, the pixel size was 0.82 Å/pixel, and the total dose was ~45 electrons/Å$^2$, distributed in 40 frames. Other data collection parameters are reported in Table S1. Urine samples were imaged at 200 kV using a Talos Arctica microscope (Thermo Fisher Scientific) with a Falcon II detector (FEI).

Movie frames were aligned using MotionCor2 (Zheng *et al*, 2017) with dose compensation, as implemented in the Scipion on-the-fly processing pipeline (de la Rosa-Trevín *et al*, 2016).

## Helical reconstruction

CTFFIND (Rohou & Grigorieff, 2015) contrast transfer function (CTF) determination, particle picking, 2D classification, 3D classification, and refinement procedures were performed using RELION (Zivanov *et al*, 2018).

For UMOD$_{fl}$, a total of 2,300 micrographs were collected and analyzed (Appendix Fig S1). ~24,000 filaments were manually picked using an assistant with enhanced visuals, and segments of 400 × 400 pixels, with 70 Å step size, were windowed out, yielding 412,322 particles for 2D classification (Appendix Fig S2A). After two rounds of 2D classifications, ~260,000 particles were chosen based on the appearance of 2D class images as well as particle image quality (rlnMaxValueProbDistribution) and resolution (rlnCtfMaxResolution) values estimated by RELION. Although most 2D classes had a distinct polarity, a few appeared to be symmetrical; the latter were thus further sub-classed in order to separate the opposite polarities (Appendix Fig S2B). Multiple 2D classes with distinct image features were identified (Appendix Fig S2C), and at first we treated these 2D classes separately. Based on the power spectrum of the 2D class images, mathematically compatible helical rise values were calculated to be 125/n, with 1<=n<=10 (Appendix Fig S2D). After testing all these possible values, it was found that only with $n = 2$ we could reproduce the angular views of any given 2D class, using the particles that belonged to it. However, reconstructed filaments originated from a single-specific 2D class appeared to be completely flat in other angular views (Appendix Fig S2E). This indicated that the distinct 2D class views might correspond to different angular views of a single type of filament with a helical rotation angle close to 180°. Thus, we pooled the 2D classes together and attempted helical reconstruction of the entire dataset. An *ab initio* low-resolution helical structure was generated using a featureless Gaussian cylinder with a diameter of 120 pixels and a length of 400 pixels as initial model. Preliminary 3D map reconstruction, however, suffered from severe flattening due to biased sampling of UMOD$_{fl}$ filament at certain rotational angles. The fact that the helical rotation angle was close to 180° made it extremely challenging to piece together a reliable initial 3D model. Extensive topological analysis was thus carried out using a ~30,000 particle subset that only included about one third of the over-represented front views, together with other angular views with stronger contrast (Appendix Figs S1 and S2F and G). This smaller dataset, which had a better angular distribution, was used to both speed up the search and minimize the effect of the intrinsically biased sampling of filaments on the cryo-grids. We exhausted all the possible center value combinations of helical parameters within the ranges 160°–200° and 60–70 Å, using allowed windows of ±5° and ±2.5 Å for each run as well as 50% overlapping search ranges between different runs. Using the same featureless Gaussian cylinder as a starting model, all helical 3D reconstruction runs converged to ~62.5 Å rise and ~180.0° rotation, except in the cases where starting helical parameters and range were too far from these values (in such cases, the parameters simply stuck at the edge of the allowed window without converging). Further helical reconstruction runs with finer step sizes were thus subsequently performed to refine the 62.5 Å/180.0° values. If the output values of the searches agreed with the middle points of the corresponding allowed windows, we halved their permitted ranges and step sizes in the following run; if they did not agree, we shifted the middle points of the allowed ranges. During this process, we also gradually included additional 2D-grouped particles that were not used during the initial helical parameter search (Appendix Fig S1). If the resulting helical 3D reconstruction runs failed to converge, we rolled back and included a smaller subset of new particles. Upon further refinement using ±0.5° and ±0.25 Å ranges, the parameters converged to a helical rise value of 62.5 Å and a helical rotation value of exactly 180.0°. After selecting the best pool of segments that shared this helical symmetry, these parameters were used for the 3D refinement of a set of 104,316 particles, which produced a 5.5 Å-resolution map of the complete filament segment. This served as a reference map (using information up to 7.0 Å resolution, in order to minimize overfitting) for additional direct 3D classification of all 412,322 particles, which captured more weakly contrasted particles with different angular views. Furthermore, additional CTF refinement and Bayesian polishing steps in combination of iterative 3D class and refinement gradually improved the maps. Finally, 288,403 particles converged into a single 3D class that ultimately yielded significantly better density after refinement with a smaller box size of 220 pixels, without helical symmetry; according to a gold standard 0.143-cutoff Fourier Shell Correlation (FSC) curve calculated using the PDBe FSC validation server (https://www.ebi.ac.uk/pdbe/emdb/validation/fsc), a resolution of 4.7 Å was reached for a central segment of density that includes five copies of the UMOD ZP module (Appendix Fig S1A and B), with the majority of the filament core having a local resolution better than 3.6 Å, accordingly to ResMap (Kucukelbir *et al*, 2014) (Appendix Fig S1C). Further improvements in map quality were obtained by either postprocessing in RELION or performing density modification with PHENIX Resolve-CryoEM (Terwilliger *et al*, 2020) without using pre-existing mask

or model information. These methods produced maps with estimated overall resolutions of 4.2 Å or 3.8 Å, respectively (Appendix Fig S1A and D), and significantly facilitated model building by both enhancing general connectivity and resolving a large fraction of protein side chains. Finally, a sharpened map of the filament core with a nominal resolution of 3.4 Å was obtained by first running ResolveCryoEM as above while specifying a smaller estimated molecular volume (based on the sequence of the EGF IV-ZP region of UMOD) and then performing iterative model-free auto-sharpening in PHENIX (Terwilliger *et al*, 2018).

When segment alignment was focused on the N-terminal region of UMOD, density corresponding to the EGF I-III and D8C domains was clearly recognizable at ~8 Å or better, without any prior mask information; this density was on the other hand absent in non-masked maps of elastase-treated UMOD samples. In order to obtain a clearer map of the N-terminal branch of $UMOD_{fl}$, we used the multi-body refinement module of RELION. From a group of particles (114,206) where the N-terminal arm was better aligned, we initiated a multi-body search and defined a single branch as a separate unit from the rest of the filament segment. Further refinement of this branch density alone produced a density with a resolution in the range of 5–7 Å, which accommodated all the branch domains of $UMOD_{fl}$ (Fig EV3C). We could identify three density patches in this map that matched well the EGF IV domain, suggesting the approximate location of EGF I-III. Based on this information, we built a molecular model of the whole N-terminal branch, and later merged this information with the high-resolution filament core structure of UMOD, in order to produce a composite map and model of $UMOD_{fl}$ (Fig EV3D).

Data processing of $UMOD_e$ was performed similarly to $UMOD_{fl}$, using 5,683 images to extract raw particles. Although $UMOD_e$ is significantly more flexible and heterogeneous than $UMOD_{fl}$, we managed to extract 252,438 usable particles (using a picking box of $350 \times 350$ pixels, with 70 Å step size) and, following a similar helical reconstruction protocol to that used for $UMOD_{fl}$ data, we determined the helical twist and rise to be $-179.9°$ and 62.7 Å, respectively. As in the case of $UMOD_{fl}$, a direct 3D classification of all particles was then performed, which identified a set of 94,937 homogeneous particles that converged into a single 3D class. 3D refinement of these particles, using a 280 Å box, fine angular sampling and no helical symmetry, produced a 6.0 Å-resolution map (FSC = 0.143 criterion). Finally, postprocessing with RELION or ResolveCryoEM yielded densities with estimated resolutions of 4.3 Å and 4.0 Å, respectively.

### Model building, refinement, validation, and analysis

Coordinates of human UMOD EGF IV/ZP-N domains (residues T296-L429) and ZP-C domain (excluding linker β-strand 1 and the EHP-containing C-terminal propeptide; residues P466-F587) were extracted from chain A of the X-ray crystallographic model of the polymerization-inhibited UMOD precursor (PDB ID 4WRN; (Bokhove *et al*, 2016a). UCSF Chimera (Pettersen *et al*, 2004) was used to first place into the central portion of the $UMOD_{fl}$ cryo-EM map a copy of the EGF IV/ZP-N fragment, whose position—despite less defined density for the relatively flexible EGF domain—was unequivocally indicated by comparison of the $UMOD_{fl}$ and $UMOD_e$ densities and consistent with a clearly corresponding elongated

region of the map (EGF IV/ZP-N(I); correlation 0.82). Subsequently, a copy of the ZP-C domain model was docked into the remaining part of the central region of the map (ZP-C(I); correlation 0.88). Additional copies of ZP-C and EGF IV/ZP-N (ZP-C(II) and EGF IV/ZP-N(II)) were then added adjacent to the previously placed EGF IV/ZP-N(I) and ZP-C(I) models, respectively, by taking into account helical symmetry information. At this stage, it became evident that an uninterrupted (and unaccounted for) extended stretch of density contacting both ZP-N(I) and ZP-C(I) linked the C-terminus of ZP-N(II) to the N-terminus of ZP-C(II) (Fig 1H and I and Movies EV2 and EV3). The latter domains, together with EGF IV(II) connected to ZP-N(II), were therefore assigned to a single molecule of UMOD (chain A, corresponding to UMOD 3 in Fig 2). Based on symmetry considerations, we also concluded that ZP-C(I) and EGF IV/ZP-N(I) belong to two distinct additional copies of UMOD (chains B and C, respectively; corresponding to UMOD 2 and UMOD 4 in Fig 2), both of which interact with chain A as well as with each other. The resulting initial set of coordinates, consisting of one chain encompassing the whole polymerization region of UMOD (A) and two half chains (B, C), was subjected to molecular dynamics (MD) flexible fitting in Namdinator (Kidmose *et al*, 2019) and manually rebuilt using Coot (Casañal *et al*, 2020) as implemented in CCP-EM (Burnley *et al*, 2017). After further improvement by Cryo_fit (Kim *et al*, 2019b), as well as additional rebuilding in Coot (whose carbohydrate-building tool (Emsley & Crispin, 2018) was used to add the N-glycan chains attached to EGF IV N322, ZP-N N396, and ZP-C N513) and ISOLDE (Croll, 2018), the model was real-space refined against the $UMOD_{fl}$ data in PHENIX (Afonine *et al*, 2018b) using a data/restraint weight of 0.8, a non-bonded weight of 250.0 and restraints generated using the starting coordinates as a reference. It was then further refined against the sharpened map of the filament core at 3.4 Å resolution, using a non-bonded weight of 250.0 and helical symmetry implicitly specified under the form of non-crystallographic symmetry constraints.

Protein and carbohydrate coordinates were validated with PHENIX (Afonine *et al*, 2018a)/MolProbity (Williams *et al*, 2018) and CCP4's Privateer (Agirre *et al*, 2015), respectively; model-to-map validation was carried out with PHENIX (Afonine *et al*, 2018a) and, in the case of sharpened maps, EMRinger (Barad *et al*, 2015). Since taken together they represent all the protein–protein interactions found in the UMOD polymer, all three UMOD chains have been included in the final deposited model, which consists of 593 protein residues (S292-F587 (chain A); S444-F587 (chain B); S292-S444 (chain C)) and 22 N-glycan residues.

The model of elastase-treated UMOD was generated by rigid body fitting the atomic coordinates of $UMOD_{fl}$ into the sharpened $UMOD_e$ map using UCSF Chimera, deleting C-terminal residues disordered due to lack of an interacting D8C domain, and carrying out refinement and validation essentially as described for $UMOD_{fl}$. The final model consists of 587 protein residues (S292-G584 (chain A); S444-G584 (chain B); S292-S444 (chain C)) and 24 N-glycan residues.

Homology modeling of the UMOD EGF I-III domain region was performed with the I-TASSER threading server (Yang & Zhang, 2015), which produced a set of coordinates that was consistent with the conserved disulfide bond pattern of other EGF domains (1–3, 2–4, 5–6) (Wouters *et al*, 2005). I-TASSER was also used to generate

models of the central region of UMOD (residues E149-S291, including the D8C domain) in parallel with *ab initio* modeling using the Robetta server (Kim *et al*, 2004). The top model produced by the latter, which had closely positioned pairs of Cys (consistent with the suggestion that all Cys in UMOD are engaged in disulfide bonds (Friedmann & Johnson, 1966; Hamlin & Fish, 1977)) and exposed to the solvent the side chains of glycosylated residues N232 and N275 (van Rooijen *et al*, 1999), was selected and refined by MD simulation in YASARA Structure (Krieger *et al*, 2009). Models combining the refined coordinates of the EGV IV/ZP module with either D8C or the whole N-terminal region of UMOD were generated by fusing in Coot molecular fragments fitted into the EM density, optimizing their fit with Namdinator and then energy minimizing the resulting coordinate sets with YASARA.

Secondary structure was assigned using STRIDE (Frishman & Argos, 1995); Poisson–Boltzmann electrostatic calculations were performed using APBS/PDB2PQR (Jurrus *et al*, 2018); protein–protein interfaces were analyzed using PISA (Krissinel & Henrick, 2007), PIC (Tina *et al*, 2007) and the AnalyseComplex command of FoldX (Delgado *et al*, 2019); intermolecular contact surface areas were calculated with dr_sasa (Ribeiro *et al*, 2019). Structural figures and movies were generated with PyMOL (Schrödinger, LLC), UCSF Chimera (Pettersen *et al*, 2004)/ChimeraX (Goddard *et al*, 2018) and Illustrate (Goodsell *et al*, 2019); unless otherwise specified, they were based on the refined coordinates of $UMOD_{fl}$.

### Sequence–structure analysis

Hidden Markov model logos were generated with Skylign (Wheeler *et al*, 2014), using as input the Pfam (El-Gebali *et al*, 2019) seed alignment for the Zona_pellucida family (PF00100), modified to include the sequence of human UMOD instead of its rat homologue and manually edited to correct a misalignment of UMOD $C_b$ to the last conserved Cys of the ZP module ($C_8$). Mapping of the evolutionary conservation of amino acid positions onto the 3D structure of UMOD was performed using ConSurf (Ashkenazy *et al*, 2016). Homology models of mouse ZP3 ZP-C and pig ZP4 ZP-N were generated using MODELLER (Webb & Sali, 2016), based on sequence–structure alignments produced by HHpred (Zimmermann *et al*, 2018).

### DNA constructs, recombinant protein expression and purification, immunoblot and immunofluorescence

Details for these methods can be found in Appendix Materials and Methods.

### Fish egg coat digest preparation

Unfertilized eggs, isolated from spawning female Japanese rice fish following procedures approved by the ethics committee of Sophia University (approval number 2016-006), were crushed in 50 mM Tris–HCl pH 7.5-buffered saline containing 5 mM ethylenediaminetetraacetic acid (EDTA) and 5 mM iodoacetic acid. After centrifugation at 2,900 *g* for 30 s, the supernatant was decanted. This procedure was repeated several times to completely remove yolk protein and cell debris. Isolated egg coats were used for digestion experiments, using HCE and LCE purified from crude hatching

liquid (Yasumasu *et al*, 1989a,b). Fifty UFEs were incubated for 1 h at 30°C in 100 ml 50 mM Tris–HCl pH 8.0 containing either 2.3 µg purified HCE or 1.2 µg purified LCE, or their mixture. The resulting material was fractionated using a HiLoad 16/60 Superdex 200 GL column (GE Healthcare) equilibrated with phosphate-buffered saline (PBS; 20 mM phosphate buffer pH 7.2, 150 mM NaCl). Two eluted protein peaks were collected and separately re-chromatographed using the same column. Final single peaks were analyzed by SDS–PAGE on 12.5% gels, as well as by native PAGE using 5-20% gradient gels.

### Cross-linking analysis of the egg envelope digest

1.75 ml egg envelope digest (~100 µg $ml^{-1}$ in PBS) was mixed with 50 µl 0.1 mg $ml^{-1}$ disuccinimidyl suberate (DSS). Cross-linking was allowed to proceed for 20 min at 25°C and quenched with the addition of 100 µl 1 M Tris–HCl pH 8.0. The protein solution was then concentrated using an Amicon Ultra 15 Ultracel-10K device (Millipore). Samples were desalted using a Millipore Ziptip C18, concentrated and analyzed with a MALDI-TOF-MS AXIMA-Performance mass spectrometer (Shimadzu). The matrix consisting of 10 mg 3,5-dimethoxy-4-hydrocinnamic acid was dissolved in a 1 ml reaction mixture containing 50% ($v/v$) acetonitrile, 0.05% ($v/v$) trifluoroacetic acid.

### Mass spectrometric analysis of the subunit composition of the egg coat digest

TOF-MS analysis of SEC-purified F2 (Fig 5A), a sample containing three protein fragments of approximately 37, 18, and 16 kDa (Fig 5B and C), showed two sharp peaks and two broad peaks (Fig 5D). Whereas the m/z values of the former (18,706.26 and 17,846.13) closely match the predicted molecular weights of ZP-C- and ZP-N-containing fragments of the ZI-1,2 subunit (18,704 (S388-G557) and 17,853 (T221-D387)), the latter (m/z 36,559.54 and 39,801.56) was assigned to a digestion product of ZI-3 that contains both ZP-N and ZP-C (34,817 (Y74-T393)) and is heterogeneously glycosylated at N184. The identity of the fragments was confirmed by Edman degradation, and a similar pattern was obtained from the analysis of SEC-purified F1 (Fig 5E).

To determine the subunit composition of F2 and F1, a cross-linking approach was employed. Each SEC peak was cross-linked by DSS, an amine-reactive compound that covalently links lysine residues, and their molecular weights were determined by MS. Cross-linking of F2 produced a single peak (Fig 5F), whose m/z 74029 matches the sum of the molecular weights of the 37, 18, and 16 kDa digestion products (73,111.93–76,353.95). Considering that F2 migrates as a single band on native PAGE (Fig 5C), this result suggests that F2 is a heterotrimeric complex containing a single copy of each digestion product. Unlike F2, DSS-cross-linked F1 consisted of several regularly spaced species, whose m/z values differed by multiples of ~18,000–20,000 (Fig 5G); this observation immediately suggests that the cross-linked complexes reflect a repeated subunit structure within the egg coat filaments. Considering that the two digestion fragments of ZI-1,2 (corresponding to the two halves of the ZP module) have an average molecular weight of ~18.3 kDa and taking into account that the species in F1 have higher molecular weights than the complex in F2 by native PAGE (Fig 5C), the MS peaks with m/z values 55605.79 and 768038.12 can be

assigned to incomplete cross-linking products consisting of half ZI-1,2 + ZI-3 and two copies of ZI-3, respectively. Similarly, higher $m/z$ peaks can be interpreted as follows: 96,658.72 = half ZI-1,2 + 2x ZI-3; 114,825.53 = 2x half ZI-1,2 + 2x ZI-3; 133,329.12 = half ZI-1,2 + 3x ZI-3; 152,398.32 = 2x half ZI-1,2 + 3x ZI-3 (Fig 5G). These assignments are consistent with the difference in intensity between the ZI-1,2 18, and 16 kDa digestion products in F1 and those in F2 (Fig 5B), and—considering that F1:F2 ratio is ~0.8 (Fig 5A and Iuchi & Yamagami, 1976)—indicate that heteromeric interactions (ZI-1,2/ZI-3) are more abundant than homomeric ones (ZI-3/ZI-3) in the UFE HCE/LCE digest.

### EM analysis of negatively stained fish egg coat filaments

After digestion with HCE to loosen filament bundles (Yasumasu *et al*, 2010) and dialysis against distilled water, unfertilized fish egg coat material (1.5 mg ml$^{-1}$) was diluted 10-fold, negatively stained with 2% uranyl acetate, and used to collect film micrographs using a Philips CM120 microscope. After digitization, 789 particles boxed from 26 filament stretches using helixboxer were used for 2D classification in EMAN2 (Tang *et al*, 2007). 32 classes were obtained, the most straight of which were used to measure distances between egg coat repeats using Fiji (Schindelin *et al*, 2012). Around 100 measurements were performed, giving an average repeat distance of ~65 Å.

### SEC-MALS analysis

An Ettan LC high-performance liquid chromatography system equipped with UV-900 detector (Amersham Pharmacia Biotech; λ = 280 nM), coupled with miniDawn Treos MALS detector (Wyatt Technology; λ = 658 nM), and an Optilab T-rEX dRI detector (Wyatt Technology; λ = 660 nM) was used to analyze the absolute molar mass of ZI proteins. Separation was performed at a flow rate of 0.5 ml min$^{-1}$, using a Superdex 200 Increase 10/300 GL column (GE Healthcare) equilibrated against 20 mM HEPES pH 7.5, 150 mM NaCl. Data processing and weight-averaged molecular mass calculations were done with Astra software (Wyatt Technology Corporation). Experiments were repeated independently twice, with each experiment including *de novo* expression, purification and measurement of ZI-1,2 or ZI-3, rather than just a repeated analysis of the same material. All measurements of each protein agreed but, for clarity, the results of a single experiment are presented in Appendix Fig S3.

### Correlation averaging analysis of UMOD filament sheets

Untilted micrographs of purified UMOD incubated with 100 mM NaCl and negatively stained with 1% uranyl acetate were subjected to image processing using the MRC program suite (Crowther *et al*, 1996). For each quasi-crystalline sheet, a Fourier filtered reference area was used to find spots in a cross-correlation function. These were employed to obtain a lattice distortion-corrected average of the repeating units. Averages from several sheets were merged following translational and rotational alignment. Analysis of Fourier amplitudes and phases extending to 1/14 Å$^{-1}$ resolution suggested the presence of two-fold symmetry perpendicular to the plane of the sheets (phase residual 25° from the theoretical 0° or 180° values); thus, p2 plane group symmetry was applied.

### Chemical depolymerization assays

Native UMOD filaments, purified as described above, were diluted to 0.25 mg ml$^{-1}$ in 0-8 M urea, 1 mM EDTA, and incubated overnight at 37°C before centrifugation at 110,000 g in a Beckman TLA 100 rotor for 1 h at 10°C. The top half of the supernatant of each sample was carefully removed for analysis, and, after discarding the remaining solution, pellets were solubilized in SDS sample buffer. Samples were analyzed on 10% SDS–PAGE gels.

## Data availability

Cryo-EM maps generated during this study have been deposited in the Electron Microscopy Data Bank (EMDB; https://wwwdev.ebi.ac.uk/pdbe/emdb) with accession codes EMD-10553 (UMOD$_{fl}$) and EMD-10554 (UMOD$_e$). Models generated during this study have been deposited in the Protein Data Bank (PDB; http://www.wwpdb.org) with accession codes 6TQK (UMOD$_{fl}$) and 6TQL (UMOD$_e$). Python scripts used for UMOD filament picking are available at https://github.com/Alexu0/Cryo-EM-filament-picking.

*Expanded View* for this article is available online.

### Acknowledgements

We thank the staff of the Swedish National Cryo-EM Facility, funded by the Knut and Alice Wallenberg Foundation, Family Erling Persson and Kempe Foundations, SciLifeLab, Stockholm University and Umeå University, for help with electron microscopes and preprocessing; the Center for High-Resolution Electron Microscopy of Karolinska Institutet for negative stain studies; the San Raffaele Advanced Light and Electron Microscopy BioImaging Center; J.M. de la Rosa-Trevín for help with Scipion; T.C. Terwilliger for advice on density modification in PHENIX; T.I. Croll for help with ISOLDE; D.S. Goodsell for advice on figure generation using Illustrate; M. Monné and S. Sandin for early experiments; and B. Forsberg for discussion. The UMOD polymerization mechanism movie was developed together with Falconieri Visuals, LLC. This work was supported by the Center for Innovative Medicine (Senior Investigator grant 2-537/2014 to L.J.), the Swedish Research Council (project grant 2016-03999 to L.J.), the Karolinska Institutet Research Foundation (grants 2016fobi50035 to L.J. and 2018-01646 to S.Z.-C.), and the Knut and Alice Wallenberg Foundation (project grant 2018.0042 to L.J.); the Ministry of Health, Singapore, NMRC grant (MOH-000382-00 to W.B.); and the Italian Ministry of Health (RF-2010-2319394 and RF-2016-02362623 to L.R.).

### Author contributions

LJ coordinated the study and designed the experiments together with SY, LR, and BW; AS, SZ-C, and LJ prepared UMOD material; AS, SZ-C, MC, and LJ collected cryo-EM data; CX, BW, AS, and LJ performed structure determination, model building, and refinement; CX wrote software for filament picking; MBr, CS, and LR produced UMOD mutants and carried out immunofluorescence analysis; SZ-C, LH, and AS expressed recombinant egg coat proteins and analyzed them by SEC-MALS; SY performed fish egg coat digestions and mass spectrometry analysis; MBo, MC, and BW analyzed fish egg coat material by EM; HH performed correlation averaging of UMOD sheets; AS and LJ wrote the manuscript with input from the other authors.

### Conflict of interest

The authors declare that they have no conflict of interest.

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
