## [Review Process File · The EMBO Journal]

Cryo-EM structure of native human uromodulin, a zona pellucida module polymer

Alena Stsiapanava, Chenrui Xu, Martina Brunati, Sara Zamora-Caballero, Céline Schaeffer, Marcel Bokhove, Ling Han, Hans Hebert, Marta Carroni, Shigeki Yasumasu, Luca Rampoldi, Bin Wu, and Luca Jovine

DOI: [10.15252/embj.2020106807](https://doi.org/10.15252/embj.2020106807)

Corresponding author(s): Luca Jovine (luca.jovine@ki.se) , Bin Wu (wubin@ntu.edu.sg)

Review Timeline:	Submission Date:	16th Sep 20
	Editorial Decision:	25th Sep 20
	Revision Received:	7th Oct 20
	Accepted:	8th Oct 20

Editor: Hartmut Vodermaier

Transaction Report:

Please note that this manuscript was previously reviewed at another journal. Since the original reviews are not subject to *The EMBO Journal's* transparent review process policy, these initial reports and author response to them cannot be published here.

Thank you again for submitting your manuscript together with previous reports and responses from another journal for our editorial consideration. As discussed earlier, we have now consulted with a trusted cryo-EM reviewer of our own journal regarding the methodological issues raised by the previous referees. Our arbitrator had access to both the latest (revised) manuscript and to the original comments and your response to them, as well as to the raw maps and model coordinates that you provided. As you will see from our expert's comments copied below, the arbitrating referee considers the density maps well sufficient for building, refining and interpreting the presented atomic models. At the same time, our referee is not convinced that the density figures as currently presented do full justice to this, and recommends (as detailed in the comments below) to revise the figures particularly in this regard. Furthermore, the arbitrator shares one specific interpretational concern of original referee 3 regarding Figure 6, and recommends that this should be presented much more cautiously (possibly also shortening this part of the currently very extensive discussion section).

In light of these overall supportive comments, we shall be happy to publish this work in The EMBO Journal, once the specific issues emphasized by our arbitrating referee have been incorporated into a final version of the manuscript. Addressing also the following important editorial points at this stage should greatly facilitate expedited consideration and processing after resubmission:

Referee #1:

The general concerns about referees 1 and 3 were about the interpretability or "buildability" of the map, in particular the so-called "density modified map". Therefore, having access to the maps and built model is indeed very helpful. Having inspected the densities, it is clear that one can properly refine an atomic model within the map. One additional consideration: as pointed out by referee 3 point 1 that the authors only docked an X-ray model. This is partially true as they needed to carry out modifications and change the connectivities in the assembled filament. In support of the author's interpretation, the fact that a large part of the model is from the X-ray structure gives better confidence in the presented model even if certain parts of the cryo-EM map are not as well resolved. This still does not diminish the presented work as it was now successfully determined in its assembled filamentous context. The authors used the relevant parts of the existing X-ray structure to restrain their building and refinement, which is good practice. The validation statistics are all within reason for that resolution. Interestingly, as you probably know in late August, another group also published the structure of the uromodulin filament (Stanisich et al., 2020). The structures agree very well (RMSD < 1 Å at this resolution fine) and this fact in itself give strong confidence that Stsiapanava also present a reliable model from the density.

My recommendation is that authors improve the figures, with a particular emphasis on convincing the reader that this map has a lot of details that are interpretable on a side-chain level, as it is commonly done for cryo-EM papers, e.g. separation of beta-strands, large bulky side chain density with an atomic model and superimposed density. This is very much missing and at the moment the reader is left wondering whether the quality of the map justifies the conclusions.

I share one additional concern about the model interpretation based on negative stain images in Figure 6 (referee 3, point 10). When left as is, it is indeed far-fetched and cannot be easily supported unless they resolve the structure. The authors should exercise more caution on this and revert to a more schematic interpretation as in Figure 7.

Response to the Referee's Comments

We thank the Referee for carefully analyzing our maps, with the conclusion that the associated atomic model has been reliably built. Here are answers to the two specific comments made by the Referee:

My recommendation is that authors improve the figures, with a particular emphasis on convincing the reader that this map has a lot of details that are interpretable on a side-chain level, as it is commonly done for cryo-EM papers, e.g. separation of beta-strands, large bulky side chain density with an atomic model and superimposed density. This is very much missing and at the moment the reader is left wondering whether the quality of the map justifies the conclusions.

We see the point of the Referee and are grateful for this suggestion. However, because the density regions shown in Fig 1 and associated Movies EV1-4 were specifically chosen to match certain areas of the structure described in the text, we thought that it would not be optimal to simply replace them. Therefore, we decided to comply with the Referee's request by generating an additional Expanded View Figure (Fig EV2) connected with Fig 1. The new figure specifically highlights the quality of the 3.4 Å map, both globally (top panel) and in detail (bottom panels):

The bottom panels provide examples of the features mentioned by the Reviewer by showing separation of β -strands (bottom left panel), as well as details of different map regions where large bulky side chains are found (bottom center and right panels).

I share one additional concern about the model interpretation based on negative stain images in Figure 6 (referee 3, point 10). When left as is, it is indeed far-fetched and cannot be easily supported unless they resolve the structure. The authors should exercise more caution on this and revert to a more schematic interpretation as in Figure 7.

We understand the concern of the Referee but, if possible, we would prefer not to replace the current UMOD sheet model representation in Fig 6 (assembled by combining multiple copies of the experimental UMOD filament map) with a more schematic depiction. This is for two different reasons. First, we think that it is important to show that the experimental map of the individual filament is in fact compatible with the proposed sheet architecture. Second, replacing the current map-based representation in Fig 6B with a schematic one (such as the one that we included in the Synopsis figure) would generate an additional issue. This is because the model depicted in Fig 6B provides an explanation for the fuzzy density that we experimentally observed in the black areas between the sheet filaments shown in the bottom panel of Figure 6A, by suggesting that this corresponds to the long glycan chains attached to N396 and N513 (arrows in Fig 6C). It is hard to envisage how to convincingly make this point, without showing a model that is based on an actual experimental map with density for such sugars. At the same time, we think that it would be even more confusing to opt for a hybrid solution where we replaced Fig 6B with a scheme but left Fig 6C (which makes an important point in favour of the model) as it is.

Since the worry here is that readers may mistakenly interpret what is shown in Figs 6B and C as an actual experimental map of the sheet (as opposed to a model of the latter, made up by assembling copies of the experimental map of a single filament), we have addressed the concern of the Referee by making the nature of the model and its limitations clearer in the legend of Fig 6. Specifically, we have modified the title of the legend so that it starts with "Modelling" ("Modelling of UMOD sheet architecture and ZP module filament binding sites for UPEC and sperm.") and edited the legend of Fig 6B so that it begins as follows:

UMOD sheet model, generated by antiparallel juxtaposition of multiple copies of the UMOD_n composite map, according to the 2D information shown in panel A. Due to lack of information in the direction perpendicular to the plane of the sheets, the exact position of adjacent filaments relative to this plane is unknown; for simplicity, the depicted model has been generated by assuming that the cores of the filaments making up the sheet lie on the same plane.

In our view, keeping the representation as it is but pointing out the above in the figure legend is a good compromise that makes the most of the experimental information that we have on the system, while at the same time making it clear to the reader that this information is incomplete in one direction and, thus, should not be confused with a real 3D model (although it

is most likely quite accurate because, in order to make a sheet, filaments must still be close enough in the perpendicular direction so that they can physically interact with each other).

Thank you for submitting your final revised manuscript for our consideration. I have now assessed your responses and modifications, and I am pleased to inform you that we have now accepted it for publication in The EMBO Journal.

Corresponding Author Name: Luca Jovine, Bin Wu

Journal Submitted to: The EMBO Journal

Manuscript Number: EMBOJ-2020-106807R